# Transcriptional repression of beige fat innervation via a YAP/TAZ-S100B axis

Xun Huang [1,2,3], Xinmeng Li[1], Hongyu Shen [1,2,3], Yiheng Zhao[1,2,3], Zhao Zhou[1], Yushuang Wang[1], Jingfei Yao[1], Kaili Xue[1], Dongmei Wu [1,2] ✉ & Yifu Qiu [1,2] ✉

Sympathetic innervation is essential for the development of functional beige fat that maintains body temperature and metabolic homeostasis, yet the molecular mechanisms controlling this innervation remain largely unknown. Here, we show that adipocyte YAP/TAZ inhibit sympathetic innervation of beige fat by transcriptional repression of neurotropic factor S100B. Adipocyte-specific loss of *Yap/Taz* induces *S100b* expression to stimulate sympathetic innervation and biogenesis of functional beige fat both in subcutaneous white adipose tissue (WAT) and browning-resistant visceral WAT. Mechanistically, YAP/TAZ compete with C/EBPβ for binding to the zinc finger-2 domain of PRDM16 to suppress *S100b* transcription, which is released by adrenergic-stimulated YAP/TAZ phosphorylation and inactivation. Importantly, *Yap/Taz* loss in adipocytes or AAV-S100B overexpression in visceral WAT restricts both age-associated and diet-induced obesity, and improves metabolic homeostasis by enhancing energy expenditure of mice. Together, our data reveal that YAP/TAZ act as a brake on the beige fat innervation by blocking PRDM16-C/EBPβ-mediated *S100b* expression.

As a dynamic metabolic organ, adipose tissue remodels its cellular size and components in response to different metabolic cues such as environmental temperature and nutritional status. One of the most metabolically beneficial remodeling is the biogenesis of beige fat. Different from classical brown fat, beige fat emerges postnatally from white adipose tissue (WAT) through a process called "browning" or "beiging" upon cold exposure or other stimuli, and exhibits strong induction of thermogenic gene program and heat production[1]. Emerging evidence suggests that recruited beige fat is associated with improvements in glucose and lipid homeostasis in ways far beyond thermogenesis[2]. Importantly, beige fat is found to exist in adult humans around supraclavicular region and displays a molecular signature resembling murine beige fat, which offers great therapeutic potential for metabolic diseases[1,3,4].

Sympathetic innervation plays a critical role in beige fat development and function[5]. Calsyntenin 3β has been proposed to promote adipocyte secretion of S100B, a neurotrophic factor, which stimulates

sympathetic innervation of themogenic fat[6]. However, a latest study reported that Calsyntenin 3β does not regulate S100B secretion and innervation but restricts lipid droplet fusion and expansion to facilitate lipid lipolysis and fatty acid oxidation[7]. Thus, it becomes unclear whether adipocyte S100B can promote sympathetic innervation and how it is regulated.

PRDM16 is a zinc finger-containing transcriptional regulator that is indispensable for beige fat biogenesis. Loss of *Prdm16* blocks the induction of a thermogenic program in cultured subcutaneous adipocytes, partially turns subcutaneous WAT (scWAT) into visceral WAT (vWAT) and impairs the recruitment of beige fat in response to β3-adrenergic or PPARγ agonists in WAT[8–10]. PRDM16 acts as a coactivator of transcriptional factors like PPARγ and PGC1α to promote thermogenic gene expression[11]. Surprisingly, little is known about how PRDM16 activity is regulated in brown and beige fat.

Both Yes-associated protein (YAP) and transcriptional coactivator with PDZ-binding motif (TAZ) are key molecular effectors of the Hippo

[1]Institute of Molecular Medicine, Beijing Key Laboratory of Cardiometabolic Molecular Medicine, College of Future Technology, Peking University, Beijing 100871, China. [2]Peking-Tsinghua Center for Life Sciences, Peking University, Beijing 100871, China. [3]Academy for Advanced Interdisciplinary Studies, Peking University, Beijing 100871, China. ✉e-mail: dongmei.wu@pku.edu.cn; yifu.qiu@pku.edu.cn

pathway in mammals, which govern cell proliferation, stem cell maintenance, and tissue homeostasis[12]. Recent studies indicate that YAP/TAZ play important roles in adipose tissue development and function. TAZ acts as a PPARγ corepressor in adipocytes to regulate cell differentiation and glucose homeostasis[13,14]. In addition, YAP/TAZ are required for adipocyte survival in diet-induced obesity and promote obesity-related adipose tissue fibrosis[15,16]. However, the roles of YAP/TAZ in beige fat development and function are unknown.

Here, we identify YAP/TAZ as corepressors of C/EBPβ by competing its binding to PRDM16. A disruption of PRDM16-C/EBPβ complex formation leads to an inhibited expression of neurotropic factor S100B. YAP/TAZ deficiency in adipocytes increases sympathetic innervation, which promotes beige fat development and thermogenesis. Importantly, targeting the YAP/TAZ-S100B axis in white adipocytes can prevent both age-associated and diet-induced obesity and ameliorates related metabolic dysfunction.

## Results

### PRDM16 interacts with YAP/TAZ

To identify regulators of PRDM16 activity in beige adipocytes, PRDM16 protein complex was immunopurified from differentiated beige adipocytes of scWAT and analyzed by mass spectrometry (MS). In accordance with previous studies, as known components of PRDM16 complex, CtBP1, CtBP2, ZFP516, and LSD1 were identified in our assay[17–19]. Of interest, we found that YAP/TAZ were pulled down in the PRDM16 complex (Fig. 1a). We confirmed the interaction between PRDM16 and YAP/TAZ via co-immunoprecipitation (co-IP) assays in 293T cell (Fig. 1b, c). Endogenous co-IP assays corroborated the interaction between PRDM16 and YAP/TAZ in beige adipocytes (Fig. 1d). We further identified that the 5th fragment of PRDM16 (F5, amino acid 880-1038), corresponding to zinc finger-2 domain (ZF2), mediates its binding to YAP and TAZ (Fig. 1e, f, h). F5 deletion abolished the interaction between PRDM16 and YAP/TAZ (Fig. 1g, h). These data demonstrate that PRDM16 interacts with YAP/TAZ through its ZF2 domain.

### Loss of *Yap/Taz* in white adipocytes promotes beige fat biogenesis

PRDM16 acts as a transcriptional coactivator to drive the development of brown adipocytes in classic BAT and to determine the beige adipocyte identity in scWAT[8,9,20–22]. Having established the interaction between PRDM16 and YAP/TAZ, we next investigated the functional implications of this interaction, particularly in adaptive thermogenesis. We first deleted *Yap* and *Taz* constitutively in mature adipocytes by crossing *Yap^{fl/fl};Taz^{fl/fl}* with *Adipoq^{Cre}* mice (*Yap^{fl/fl};Taz^{fl/fl};Adipoq^{Cre}*, termed YT-AKO) and performed bulk RNA sequencing (RNA-seq) analysis of scWAT from YT-AKO and control mice housed at room temperature. Immunoblot analysis confirmed the knockout efficiency as shown by an evident decrease of YAP/TAZ protein expression both in scWAT and BAT from YT-AKO mice (Fig. 2d and Supplementary Fig. 1d). Interestingly, KEGG and BP (Biological Process) pathway analyses showed that metabolic pathways, mitochondrial oxidative phosphorylation and thermogenesis pathways were enriched on the top among the upregulated after deletion of *Yap/Taz* (Fig. 2a). Besides, neurodegenerative disease (Huntington, Parkinson and Alzheimer) pathways were also enriched, whereas the gene sets of them were nearly same with that of thermogenesis pathway (Supplementary Fig. 1a). Together, this Gene Ontology (GO) analysis indicates that thermogenesis pathway was greatly activated in scWAT from YT-AKO mice. Indeed, nearly all the genes in the set of thermogenesis pathway were upregulated in YT-AKO group (Fig. 2b). To validate the RNA-seq results, we performed RT-qPCR and immunoblot analyses and found that loss of *Yap/Taz* in scWAT resulted in a robust induction of thermogenic program like *Ucp1* and *Elovl3* (Fig. 2c, d) but did not influence expression of adipocyte identity gene (Supplementary Fig. 1b).

Hematoxylin and Eosin (H&E) staining analysis displayed that YT-AKO scWAT had more beige fat both under room temperature and cold exposure conditions (Fig. 2e). These data indicate that loss of *Yap/Taz* promotes beige fat biogenesis in scWAT in an adipogenesis-independent manner. However, the expression of thermogenic program was not altered in YT-AKO BAT (Supplementary Fig. 1c, d). To further confirm this, we crossed *Yap^{fl/fl};Taz^{fl/fl}* with *Ucp1^{Cre}* mice (*Yap^{fl/fl};Taz^{fl/fl};Ucp1^{Cre}*, termed YT-UKO) to specifically delete *Yap/Taz* in BAT. Immunoblot data revealed no change of UCP1 expression in BAT after deletion of *Yap/Taz* (Supplementary Fig. 1e). Interestingly, the YT-UKO mice showed an abnormal circling behavior, suggesting a nervous system dysfunction (Supplementary Movie 1). Consistent with the increased thermogenic gene expression in scWAT, higher oxygen consumption and lower respiratory exchange rate (RER) were observed in YT-AKO mice (Fig. 2f–h), indicating higher energy expenditure and fatty acid oxidation. Accordingly, core body temperature of YT-AKO mice was significantly higher than control mice upon cold exposure (Fig. 2i, j). Furthermore, YT-AKO mice showed higher oxygen consumption and lower RER upon a stimulation by CL-316,243, a β3-adrenoceptor agonist, excluding a potential contribution of shivering thermogenesis under cold exposure condition (Fig. 2k, l).

To exclude potential systemic effects of *Adipoq^{Cre}* mouse model, we used our recently-developed method to specifically delete *Yap/Taz* in mature adipocytes by local injection of AAV-ADP-Cre in *Yap^{fl/fl};Taz^{fl/fl}* scWAT[16]. RT-qPCR and immunoblot assays revealed that thermogenic genes were dramatically induced in scWAT after deleting *Yap/Taz* (Supplementary Fig. 1f, g). To exclude potential developmental issues of constitutive YT-AKO mouse model, we employed inducible adipocyte-specific *Yap/Taz*-deficient mice by crossing *Yap^{fl/fl};Taz^{fl/fl}* with *Adipoq^{CreERT2}* mice (*Yap^{fl/fl};Taz^{fl/fl};Adipoq^{CreERT2}*, termed YT-iAKO). Indeed, YT-iAKO mice also showed higher expression of thermogenic program like UCP1 in scWAT after tamoxifen administration (Supplementary Fig. 1h, i). While *Yap/Taz* double knockout in mature adipocytes induced thermogenic program, no such alteration could be observed in adipocyte-specific *Yap* or *Taz* single knockout mice (Y-AKO or T-AKO) (Supplementary Fig. 1j, k), indicating a redundant role of YAP and TAZ in the regulation of beige fat biogenesis. Together, these results demonstrate that loss of *Yap/Taz* in mature adipocytes promotes beige fat development and thermogenesis.

### *Yap/Taz* deficiency increases sympathetic innervation by inducing S100B expression

To define the mechanism by which *Yap/Taz* loss induces beige fat biogenesis, we simultaneously deleted *Yap* and *Taz* in differentiated beige adipocytes using CRISPR-Cas9 method with tandem *Yap/Taz*-gRNAs[16] and found no change of thermogenic and adipocyte marker gene expression (Supplementary Fig. 2a–d). Moreover, YAP/TAZ loss in differentiated beige adipocytes did not alter adipocyte identity as revealed by equal lipid accumulation (Supplementary Fig. 2e, f). These data indicate that YAP/TAZ deficiency induces beiging in a non-adipocyte-autonomous manner and other cellular components in adipose tissue may mediate this effect. Since sympathetic nervous system is critical for beige adipogenesis, we used Adipo-Clear and 3D adipose tissue imaging methods to visualize sympathetic innervation[23,24] and found that scWAT from YT-AKO mice exhibited much denser sympathetic innervation particularly parenchymal one (Fig. 3a, b, Supplementary Fig. 2g and Supplementary Movie 2). Consequently, there was more UCP1 signal in scWAT from YT-AKO mice (Supplementary Fig. 2h). To understand how sympathetic innervation was increased, we reanalyzed the RNA-seq data of scWAT from YT-AKO mice and found that *Ucp1* and *S100b* were strikingly induced and ranked on the top in the upregulated group (Fig. 3c). S100B was reported to be mainly expressed by astrocytes in the central nervous system and adipocytes in fat pad, functioning as a neurotoxic or neuroprotective factor[6,25]. Immunoblot and RT-qPCR analyses

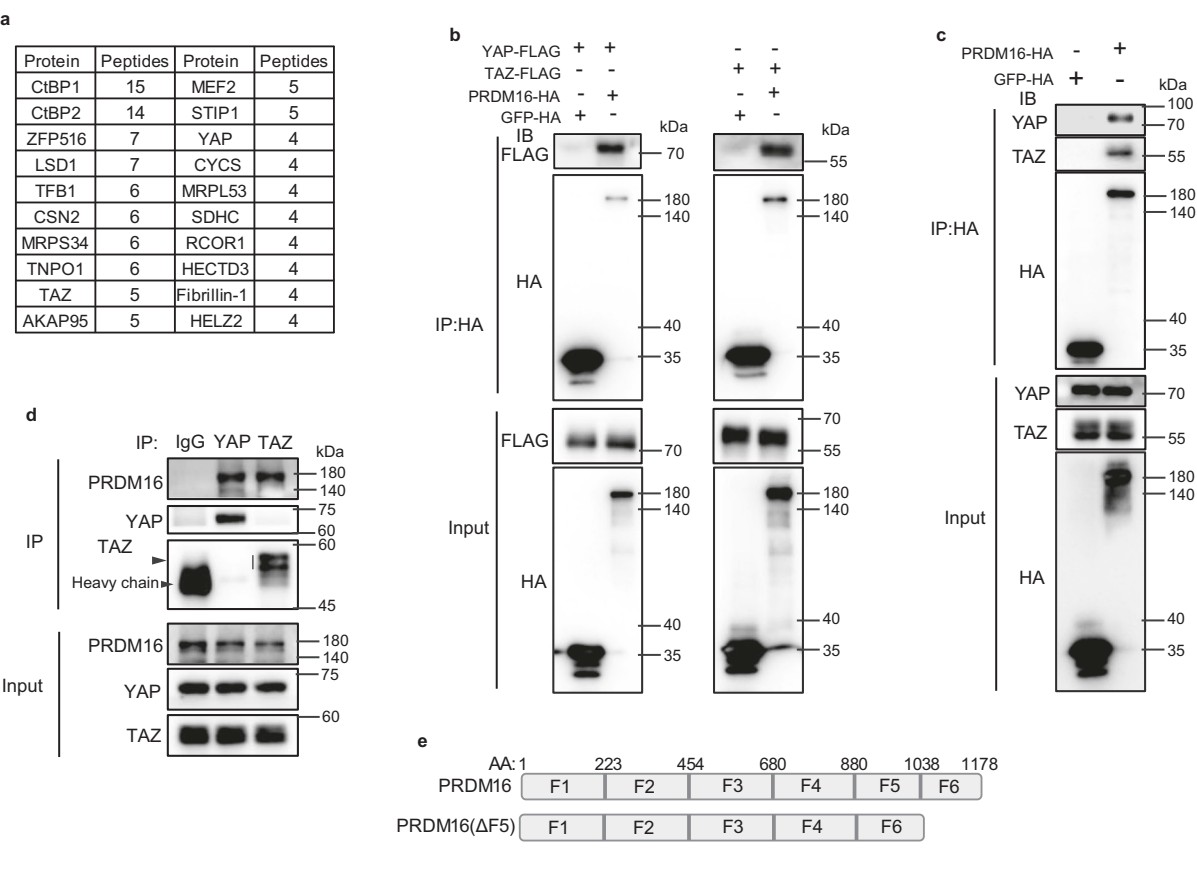

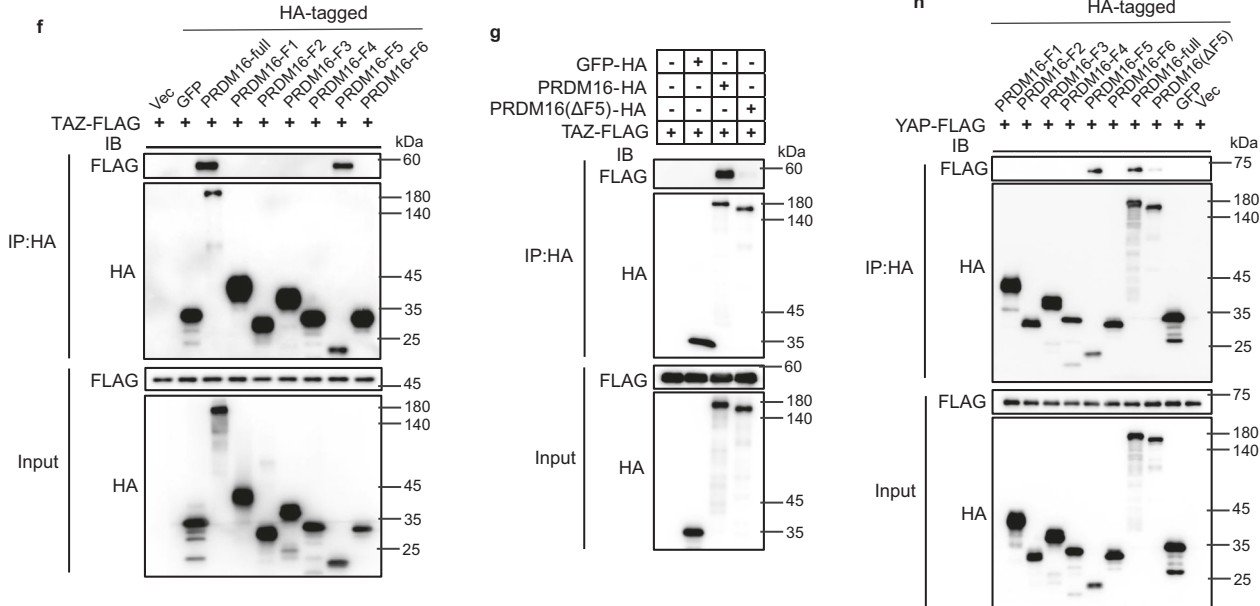

**Fig. 1 | PRDM16 interacts with YAP and TAZ through ZF2 domain. a** PRDM16-associated proteins identified by LC-MS/MS (top 20 according to the total numbers of peptides identified). **b** Co-IP of overexpressed PRDM16 and YAP/TAZ in HEK-293T cells. IB, immunoblot; co-IP, co-immunoprecipitation. **c** Co-IP of over-expressed PRDM16 and endogenous YAP/TAZ in HEK-293T cells. **d** Co-IP of endogenous YAP/TAZ and PRDM16 in SVF-derived beige cells. **e** Schematic of PRDM16 fragments and PRDM16(ΔF5). F5 is ZF2 domain. AA, amino acid. **f** Co-IP of TAZ and PRDM16 fragments in HEK-293T cells. **g** Co-IP of TAZ and PRDM16(ΔF5) in HEK-293T cells. **h** Co-IP of YAP and PRDM16 fragments in HEK-293T cells. For (**b**–**d**) and (**f**–**h**), representative results are from two independent experiments. Source data are provided as a Source data file.

confirmed the robust induction of *S100b* in YT-AKO scWAT (Fig. 3d and Supplementary Fig. 3a). Besides S100B, other neuregulins, including nerve growth factor (NGF), brain-derived neurotrophic factor (BDNF), neurotrophin-3(NTF3), neurotrophin-4 (NTF4) and

neuronal growth regulator 1 (NEGR1), have been implicated in sympathetic innervation of thermogenic fat[26–30]. We thus examined all these factors in scWAT and found that only *S100b* was induced upon *Yap/Taz* deletion (Supplementary Fig. 3a). Moreover, RNA-seq data

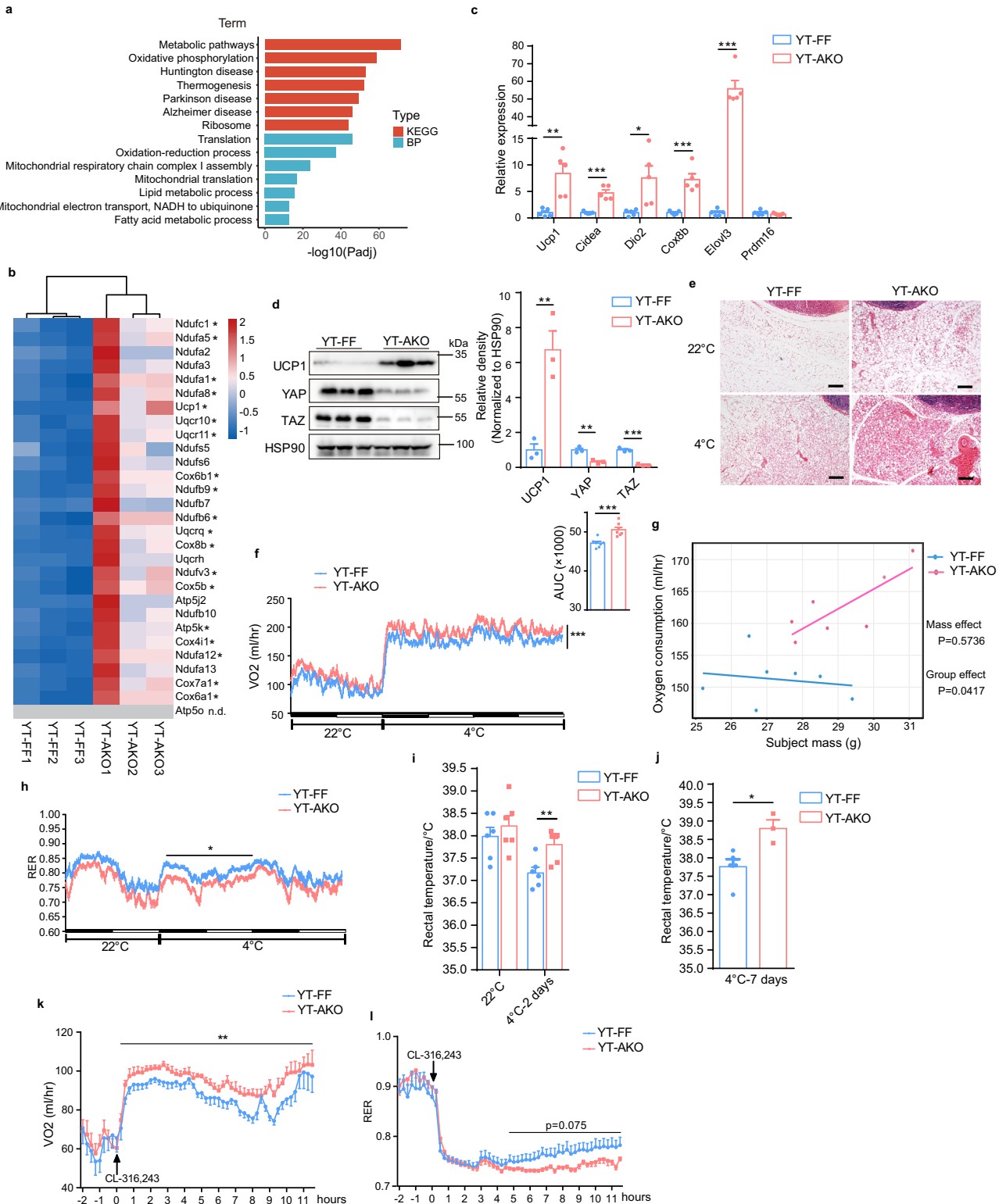

**Fig. 2 | Adipocyte-specific *Yap/Taz* deletion induces beige fat biogenesis and thermogenesis in scWAT. a** Gene Ontology (GO) enrichment analysis of significantly upregulated genes (log₂FC > 1, Padj < 0.01) in scWAT after deletion of *Yap/Taz*. **b** Heat map of thermogenesis gene set as in (**a**). n.d., not detectable. *Padj < 0.05. **c** Relative mRNA levels of thermogenic genes in scWAT. **d** Immunoblot and quantification of indicated protein in scWAT as in (**c**). **e** H&E sections of scWAT from mice housed at 22 °C and 4 °C, respectively. Scale bars: 100 μm. **f, g** Oxygen consumption of YT-AKO and YT-FF mice. AUC, area under curve. **h** Respiratory exchange ratio (RER) as in (**f**). **i, j** Core body temperature of mice after cold

exposure for 2 days (**i**) or 7 days (**j**). **k, l** Oxygen consumption (**k**) and RER (**l**) of mice administrated with CL-316,243. *N* = 3 (**b, d, J**−YT-AKO), 5 (**c, J**−YT-FF), 6 (**i**), 7 (**f**−**h**) or 9 mice (**k, l**) per group, and data are mean ± s.e.m. Benjamini−Hochberg-adjusted one-sided hypergeometric test (**a**); Benjamini−Hochberg-adjusted two-tailed unpaired Student's *t*-test (**b**); two-tailed unpaired Student's *t*-test (**c, d, f**−AUC, **i, j**); two-way analysis of variance (ANOVA) (**f, h, k, l**); CalR-ANCOVA (**g**). *$P < 0.05$, **$P < 0.01$, ***$P < 0.001$. Specific *p*-values and source data are provided as a Source data file.

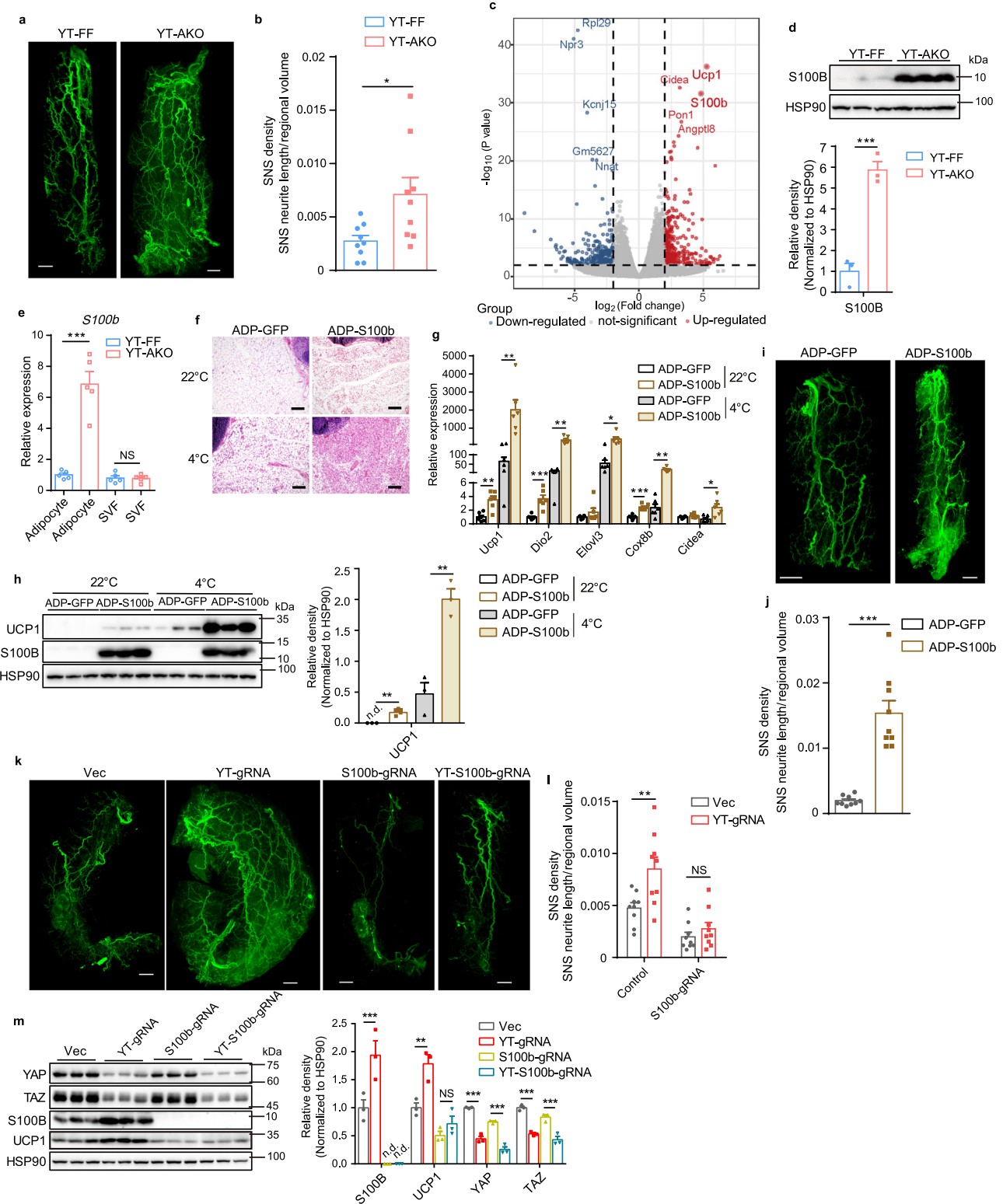

revealed that the expression of *S100b* is much higher than other factors in scWAT (Supplementary Fig. 3b). In order to identify which cell type contributed to the increased S100B expression in scWAT, we separated scWAT into mature adipocytes and stromal vascular fraction (SVF) and found that *S100b* was specifically induced in mature adipocytes from YT-AKO mice (Fig. 3e). Loss of *Yap/Taz* in differentiated beige adipocytes induced the expression of *S100b* significantly (Supplementary Fig. 3c, d), indicating that YAP/TAZ regulate *S100b* expression in a cell-autonomous manner.

To determine the functional implications of *Yap/Taz* loss-induced *S100b* expression, we specifically overexpressed *S100b* in adipocytes for 4 weeks by locally delivering AAV-ADP-*S100b* in scWAT. We found that ectopically-expressed S100B robustly induced beige fat biogenesis in scWAT especially after cold exposure (Fig. 3f). Consistently, the expression of thermogenic genes such as *Ucp1*, *Dio2*, *Cox8b*, and *Elovl3* was drastically upregulated in scWAT before and especially after cold exposure (Fig. 3g, h), while no change was observed in adipocyte identity gene expression (Supplementary Fig. 3e). When

**Fig. 3 | YAP/TAZ deficiency promotes S100B expression in adipocytes to stimulate sympathetic innervation. a, b** Whole-mount TH immunostaining (**a**) and quantification (**b**) of cleared scWAT from YT-AKO and YT-FF mice. Scale bars: 1 mm. **c** A volcano plot of changed genes measured by RNA-seq in scWAT from YT-AKO and YT-FF mice. The highlighted are genes whose expression were significantly altered (fold change>4, *P* < 0.01). **d** Immunoblot and quantification of S100B in scWAT as in (**c**). **e** Relative mRNA level of *S100b* in mature adipocytes and SVF from scWAT as in (**c**). **f** H&E sections of scWAT from wild-type mice ectopically expressed with S100B and GFP under 22 °C or 4 °C for 2 days. Scale bars: 100 μm. **g** Relative mRNA levels of thermogenic genes in scWAT as in (**f**). **h** Immunoblot and quantification of indicated proteins in scWAT as in (**f**). **i, j,** Whole-mount TH immunostaining (**i**) and quantification (**j**) of cleared scWAT as in (**f**) (22 °C). Scale bars: 1 mm. **k, l** Whole-mount TH immunostaining (**k**) and quantification (**l**) of cleared scWAT in *Rosa26-LSL-Cas9;Adipoq^Cre^* scWAT injected with AAV-Vec, AAV-*Yap/Taz*-gRNA (YT-gRNA), AAV-*S100b*-gRNA, AAV-*Yap/Taz-S100b*-gRNA (YT-S100b-gRNA), respectively. Scale bars: 1 mm. **m** Immunoblot and quantification of indicated proteins of scWAT as in (**k**). n.d., not detectable. *N* = 9 random subregions from 3 independent tissues per group (**b, j, l**). *N* = 3 (**d, h, m**), 5 (**e**) or 6 (**g**) mice per group, and data are mean ± s.e.m. Two-tailed unpaired Student's *t*-test (**b, d, e, g, h, j**); Wald test (**c**); two-way ANOVA with Tukey's multiple comparisons test (**l, m**). **P* < 0.05, ***P* < 0.01, ****P* < 0.001; NS, not significant. Specific p-values and source data are provided as a Source data file.

overexpressing S100B in differentiated beige adipocytes, thermogenic *Ucp1* and adipogenic genes were not altered (Supplementary Fig. 3f, g), further supporting that S100B promotes beige fat biogenesis through a non-adipocyte-autonomous mechanism. To examine whether S100B is responsible for *Yap/Taz* loss-induced sympathetic innervation increase of scWAT, we performed S100B overexpression in scWAT of wild-type mice and *S100b/Yap/Taz* knockout by local injection of respective AAV-tandem gRNAs in scWAT of *Rosa26-LSL-Cas9;Adipoq^Cre^* mice[31]. The results show that enforced expression of S100B in scWAT profoundly promoted sympathetic parenchymal innervation (Fig. 3i, j and Supplementary Fig. 3h). Consistently, *Yap/Taz* deletion increased sympathetic innervation in scWAT, which was blocked when deleting *S100b* simultaneously (Fig. 3k, l). Also, *Yap/Taz* loss-induced UCP1 expression in scWAT disappeared when deleting *S100b* meanwhile (Fig. 3m). *Yap/Taz* loss in scWAT bona fide induced *S100b* expression not only in YT-AKO mice but also in YT-iAKO mice and in *Yap^fl/fl^/Taz^fl/fl^* mice locally injected with AAV-ADP-Cre (Supplementary Fig. 3i, j). Besides male mice, we also repeated the experiments in adult female mice and got similar results (Supplementary Fig. 3k, l). Together, these findings demonstrate that *Yap/Taz* loss in adipocytes induces expression of *S100b* that increases sympathetic innervation and thus beige fat biogenesis in scWAT.

## YAP/TAZ impede C/EBPβ-PRDM16 complex formation to suppress *S100b* expression

Since YAP/TAZ interact with PRDM16 that promotes *S100b* expression by an unknown mechanism[6], we hypothesized that YAP/TAZ regulates *S100b* expression by regulating PRDM16's activity. Through analyzing the ChIP-seq data of PRDM16 (GSE63965), we found that PRDM16 can bind to the promoter of *S100b* (SP), which was validated by Cut&Tag-qPCR assay (Fig. 4a, h). Whole-body knockout or transgenic overexpression of PRDM16, respectively, decrease or increase *S100b* expression in scWAT[6]. To exclude potential systemic effects of whole-body knockout on *S100b* expression, we locally delivered AAV-*Prdm16*-gRNA into scWAT of *Rosa26-LSL-Cas9;Adipoq^Cre^* mice to knock out *Prdm16* specifically in adipocytes. RT-qPCR results showed that *Prdm16* knockout in scWAT adipocytes decreased *S100b* expression (Supplementary Fig. 4a, b). On the other hand, forced expression of PRDM16 in differentiated beige adipocytes markedly promoted *S100b* expression, which was largely blocked by co-expression of YAP/TAZ (Supplementary Fig. 4c, d). Together, these results reveled that PRDM16 promotes *S100b* expression by binding to the SP in a cell-autonomous manner, which can be blocked by YAP/TAZ.

We next examined whether PRDM16 directly bind to the promoter of *S100b* to regulate its transcription as PRDM16 can directly bind to DNA[32]. Forced expression of PRDM16 (R998Q) mutant, which loses DNA-binding capacity[20], promoted *S100b* expression in differentiated beige adipocytes to a comparable level of wild-type PRDM16, suggesting that PRDM16 promotes *S100b* expression through collaborating with other transcriptional factors (TFs) but not directly binding to the promoter (Supplementary Fig. 4e, f). To identify such TFs, we used a TF-binding prediction tool to predict potential SP-binding TFs and got forty-nine potential TFs (Fig. 4b). Using LC-MS/MS approach,

Yutaka et al. identified forty-four potential DNA-binding TFs in the immunopurified PRDM16 complex from the nuclei of differentiated beige adipocytes[33]. We crossed these two TF lists and got three common hits that are C/EBPα, C/EBPβ, and PPARγ (Fig. 4b). To verify which TF mediates PRDM16's activity, we individually expressed them in differentiated beige adipocytes and found that only C/EBPβ could robustly promote *S100b* expression (Fig. 4c and Supplementary Fig. 4g). C/EBPβ is essential for adipocyte fate determination and adipogenesis, and forms a transcriptional complex with PRDM16 to initiate a switch from myoblast to brown fat[22,34]. We thus speculated that PRDM16 interacts with C/EBPβ that binds to the SP to promote *S100b* transcription. Through analyzing the ChIP-seq data of C/EBPβ performed in 3T3-L1 cells (GSM686970), we found that C/EBPβ binds to the SP (Fig. 4d). However, C/EBPα, another member of C/EBP family, did not bind to the SP according to the ChIP-seq data performed in brown adipocytes (GSM1218859) (Supplementary Fig. 4h). Using Cut&Tag-qPCR method, we validated that C/EBPβ abundantly bond to the SP in differentiated beige adipocytes (Fig. 4e). When co-expressing C/EBPβ and PRDM16, the *S100b*-inducing effect was much stronger than C/EBPβ or PRDM16 alone, suggesting that they are working in a synergistic way (Fig. 4f and Supplementary Fig. 4i). However, when expressing YAP or TAZ simultaneously, the *S100b*-inducing effect by the PRDM16-C/EBPβ complex disappeared (Fig. 4f and Supplementary Fig. 4i).

Then the next question is how YAP/TAZ negatively regulate this process. We found that *Yap/Taz* deletion in adipocytes did not influence the expression of PRDM16 and C/EBPβ (Supplementary Fig. 4j, k), indicating that other mechanisms are involved rather than YAP/TAZ regulation of PRDM16 and C/EBPβ expression. Given that PRDM16 interacts with YAP, TAZ, and C/EBPβ all through its ZF2 domain (Fig. 1)[22], we speculated that YAP/TAZ compete with C/EBPβ for ZF2 domain binding and thus impede PRDM16-C/EBPβ complex formation. Indeed, co-IP assays in 293 T cells showed a marked blockade of PRDM16-C/EBPβ complex formation by YAP and TAZ (Fig. 4g). Consistently, Cut&Tag-qPCR results reveled that YAP/TAZ blocked PRDM16's binding to SP (Fig. 4h). To determine whether the PRDM16-C/EBPβ complex mediates *Yap/Taz* loss-induced *S100b* expression, we used AAV-tandem gRNA to delete *Yap*, *Taz* and *Prdm16* or *Cebpb* combinatorially in beige adipocytes and found that *Yap/Taz* loss-induced *S100b* expression disappeared once *Prdm16* or *Cebpb* was simultaneously deleted (Fig. 4i and Supplementary Fig. 4l). Besides, we isolated scWAT from adult YT-AKO and control mice, and co-IP assays revealed that YAP/TAZ deficiency could significantly enhance the formation of PRDM16-C/EBPβ complex (Fig. 4j). However, YAP/TAZ loss did not influence PRDM16-C/EBPβ complex formation in BAT (Supplementary Fig. 4m). Since cold exposure induces *S100b* expression[6], we thus investigated whether C/EBPβ get involved in this process. We found that adipocyte-specific expression of C/EBPβ by local injection of AAV-ADP-*Cebpb* in scWAT robustly enhanced *S100b* and *Ucp1* expression after cold exposure (Fig. 4k and Supplementary Fig. 4n, o). To evaluate whether this regulation axis exists in preadipocytes, we examined the expression of PRDM16, C/EBPβ, YAP, and TAZ in

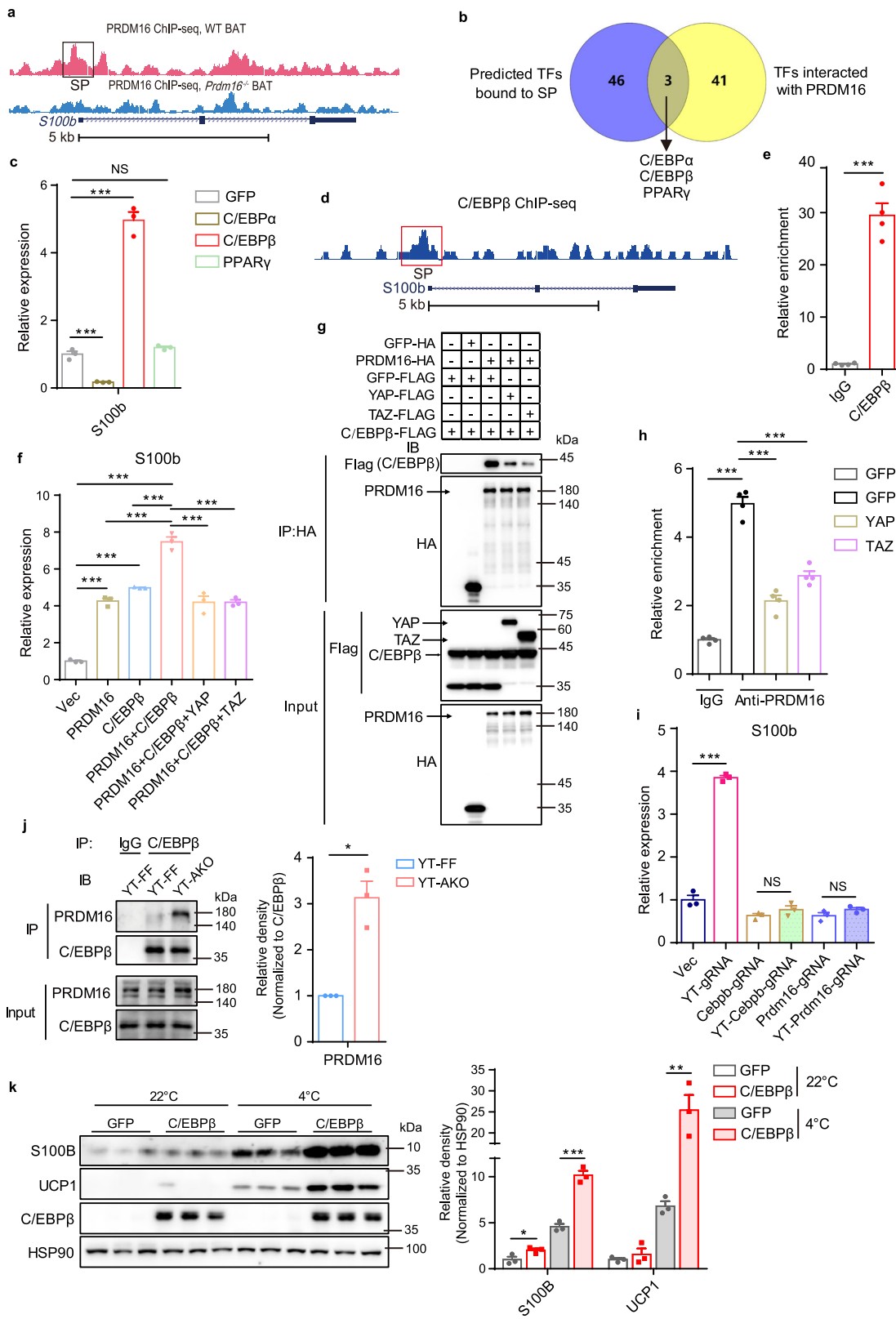

preadipocytes and beige adipocytes and found that all these proteins especially PRDM16 and YAP were highly expressed in adipocytes, suggesting that the YAP/TAZ-PRDM16-C/EBPβ axis mainly operates in differentiated beige adipocytes (Supplementary Fig. 4p). Together, these data indicate that YAP/TAZ disrupt PRDM16-C/EBPβ complex formation to inhibit *S100b* expression in adipocytes.

**Cold-induced adrenergic signaling inactivates YAP/TAZ to derepress S100B expression and sympathetic innervation**

YAP/TAZ are core transcriptional coregulators of the Hippo pathway, and they bind to transcriptional factors TEADs (TEAD1–4) to regulate target genes' expression[35]. To determine whether TEADs are involved in the regulation of *S100b* expression, we expressed TEADs binding-deficient mutants of YAP (S79A) and TAZ (S51A)[36] in differentiated

**Fig. 4 | YAP/TAZ suppress C/EBPβ-PRDM16 complex formation to inhibit S100b expression. a** Alignment of PRDM16 ChIP-seq peaks at *S100b* gene locus in BAT. *S100b* gene is structured with introns (line), exons (box), and direction (arrow). **b** A Venn diagram showing the putative transcriptional factors (TFs) bound to *S100b* promoter (SP) and interacted with PRDM16. **c** Relative mRNA levels of *S100b* in SVF-derived beige adipocytes expressed with indicated proteins. **d** Alignment of C/EBPβ ChIP-seq peaks at *S100b* gene locus in 3T3-L1 cells. **e** Relative enrichment of C/EBPβ on SP in SVF-derived beige adipocytes. **f** Relative mRNA levels of *S100b* in SVF-derived beige adipocytes expressed with indicated proteins. **g** Co-IP of C/EBPβ and PRDM16 in HEK-293T cells expressed with indicated proteins. **h** Relative enrichment of PRDM16 on SP in SVF-derived beige adipocytes expressed with indicated proteins. **i** Relative mRNA levels of *S100b* in SVF-derived beige adipocytes from

Cas9[tg/tg] scWAT injected with AAV-Vec, AAV-*Yap/Taz*-gRNA, AAV-*Cebpb*-gRNA, AAV-*Yap/Ta*z-*Cebpb*-gRNA, AAV-*Prdm16*-gRNA, AAV-*Yap/Taz-Prdm16*-gRNA, respectively. **j** Endogenous co-IP of PRDM16 and C/EBPβ and quantification of immunoprecipitated PRDM16 in scWAT from YT-AKO and YT-FF adult mice (*n* = 3 per group). **k** Immunoblot and quantification of indicated proteins in scWAT injected with AAV-ADP-C/EBPβ or AAV-ADP-GFP for 4 weeks under 22 °C and 4 °C for 2 days (*n* = 3 mice per group). *N* = 3 (**c, f, i**) or 4 (**e, h**) biologically independent cell cultures per group, and data are mean ± s.e.m. Two-tailed unpaired Student's *t*-test (**e, j, k, h**: IgG-GFP versus anti-PRDM16-GFP group); one-way ANOVA with Bonferroni's multiple comparisons test (**c, f, h**: anti-PRDM16 groups); two-way ANOVA with Tukey's multiple comparisons test (**i**). *$P < 0.05$, **$P < 0.01$, ***$P < 0.001$; NS, not significant. Specific *p*-values and source data are provided as a Source data file.

beige adipocytes and found that they could still inhibit *S100b* expression to similar extents as wild-type YAP/TAZ (Supplementary Fig. 5a). In supporting of this point, co-IP assays showed that YAP (S79A) and TAZ (S51A) disrupted PRDM16-C/EBPβ complex to a similar extent as wild-type YAP/TAZ (Supplementary Fig. 5b, c). These data indicate that YAP/TAZ regulate S100b expression in a TEADs-binding-independent manner.

Previous studies have shown that cold exposure can promote both S100B expression and sympathetic innervation in scWAT[6,24], yet the underlying mechanism is still unknown. Cold exposure induces noradrenaline (NE) secretion to activate β-adrenergic receptors (β-AR), which in turn induces cyclic adenosine monophosphate (cAMP) accumulation and consequently protein kinase A (PKA) activation[37]. This adrenergic signaling plays a dominant role in adipocyte thermogenesis and lipolysis under cold condition[38]. We thus speculated that the NE-β-AR-cAMP-PKA signaling may be responsible for cold-induced S100B expression. Indeed, both NE and FSK (an adenylyl cyclase activator) effectively promoted S100B expression in beige adipocytes, which was blocked by H89 (a PKA inhibitor) (Fig. 5a). To further explore the role of YAP/TAZ during this process, we found that NE-stimulated phosphorylation of YAP (pYAP-S112) and TAZ (pTAZ-S89) in beige adipocytes, peaking at 30 min and then gradually attenuating (Fig. 5b). Also, we observed that FSK robustly induced YAP/TAZ phosphorylation while H89 abolished NE-induced YAP/TAZ phosphorylation, indicating that the cAMP-PKA axis mediates this induction (Fig. 5c). This is consistent with a previous study showing that PKA regulates cell proliferation and differentiation by regulating Hippo pathway activaty[39]. Furthermore, β3-adrenergic receptor agonist CL-316,243 simulated YAP/TAZ phosphorylation and promoted *S100b* expression as well (Supplementary Fig. 5d, e).

We next investigated how the increased YAP/TAZ phosphorylation regulates *S100b* expression. Given that YAP/TAZ phosphorylation promotes their nucleus-to-cytoplasm shuttling, we first examined their subcellular abundance by immunofluorescence and found that YAP/TAZ were decreased in the nucleus while PRDM16 was not altered upon NE stimulation or PKA activation (Fig. 5d–g and Supplementary Fig. 5f), which was confirmed by nuclear and cytoplasmic protein extraction assays (Supplementary Fig. 5g–i). Consistent with lower level of YAP/TAZ in nucleus upon NE or FSK treatment, the formation of PRDM16-C/EBPβ complex was increased (Fig. 5h). In addition, nucleus-localized YAP (5SA) and TAZ (4SA) could block PRDM16-C/EBPβ formation more robustly than wild-type YAP and TAZ (Supplementary Fig. 6a, b). These findings indicate that the NE-cAMP-PKA signaling induces YAP/TAZ phosphorylation that excludes them from the nucleus and thus separate them from and release their inhibition on the formation of nuclear PRDM16-C/EBPβ complex. We corroborated this conclusion in vivo by showing that cold exposure stimulated YAP/TAZ phosphorylation in scWAT (Supplementary Fig. 6c, d). Furthermore, cold exposure could not induce S100B expression anymore and UCP1 induction was also greatly decreased in scWAT when expressing TAZ (4SA) in scWAT[40] (Fig. 5i and Supplementary Fig. 6e, f). In the meanwhile, cold exposure-induced sympathetic innervation was greatly impaired in scWAT

expressed with TAZ (4SA) (Fig. 5j, k). These effects were not due to impaired adipocyte development because TAZ (4SA) expression had not affected expression of adipocyte identity genes (Supplementary Fig. 6g). Together, these findings reveal that cold-induced adrenergic signaling promotes YAP/TAZ phosphorylation and sequester them in the cytoplasm to release their binding to and inhibition on nuclear PRDM16, which promotes S100B expression and subsequently sympathetic innervation.

### *Yap/Taz* loss in adipocytes protects mice from age-associated and HFD-induced obesity

Both in humans and mice, aging is linked to increased fat mass, loss of lean mass, impaired insulin sensitivity and decreased energy expenditure[41,42]. Aging is also associated with white adipose tissue neuropathy, leading to a loss of innervation around tissue vasculature[43]. These observations prompted us to ask whether YAP/TAZ deficiency could improve WAT sympathetic innervation and energy homeostasis in aging condition. Insulin tolerance test and glucose tolerance test revealed that old YT-AKO mice had improved insulin sensitivity and glucose clearance capacity compared to control mice (Fig. 6a, b), while there was no difference when they were young (Supplementary Fig. 7a, b). Although the old YT-AKO mice had same body weight as their controls, their fat mass was markedly decreased while their lean mass was increased (Fig. 6c). Specifically, the size and weight of white adipose tissue including scWAT and epididymal WAT (eWAT) but not BAT were smaller and lighter than controls (Fig. 6d, e), although lipolytic ability did not change between YT-AKO and control mice as revealed by similar phosphorylation level of Hormone-sensitive lipase (HSL) and Perilipin1 (Supplementary Fig. 7c–f). Accordingly, YT-AKO mice had smaller adipocytes in scWAT and eWAT, which might contribute to the improved insulin sensitivity (Fig. 6f, g). Consistent with greatly upregulated expression of thermogenic genes like *Ucp1*, energy expenditure (VO2) of YT-AKO mice was 13.7% higher than that of controls (Fig. 6h–k), with no difference in food intake and locomotor activity (Supplementary Fig. 7g, h). Considering a large reduction of lipid content but no liver steatosis in YT-AKO old mice, we measured serum level of NEFA and TG and found that no more lipid metabolites accumulated, suggesting that the lost fat was burned by beige fat (Supplementary Fig. 7i, j). Mechanistically, the expression of neurotrophic factor S100B and tyrosine hydroxylase (TH, a marker of sympathetic nerves) were largely increased in YT-AKO scWAT (Fig. 6j, k). Whole-mount imaging also revealed a denser sympathetic innervation in YT-AKO scWAT (Fig. 6l, m). Together, these results indicate that loss of *Yap/Taz* in adipocytes improves systemic metabolism of old mice through inducing beige fat biogenesis.

Our previous study revealed that WAT from obese mice expresses higher level of YAP/TAZ[16], which suggested that targeting YAP/TAZ in adipose tissue may improve systemic metabolism of obesity. When fed high-fat diet (HFD), YT-AKO mice gained less body weight starting at 7 weeks of HFD feeding and maintained at similar level as control mice fed normal diet (ND) (Supplementary Fig. 8a). This obesity-resistant phenotype of YT-AKO mice was

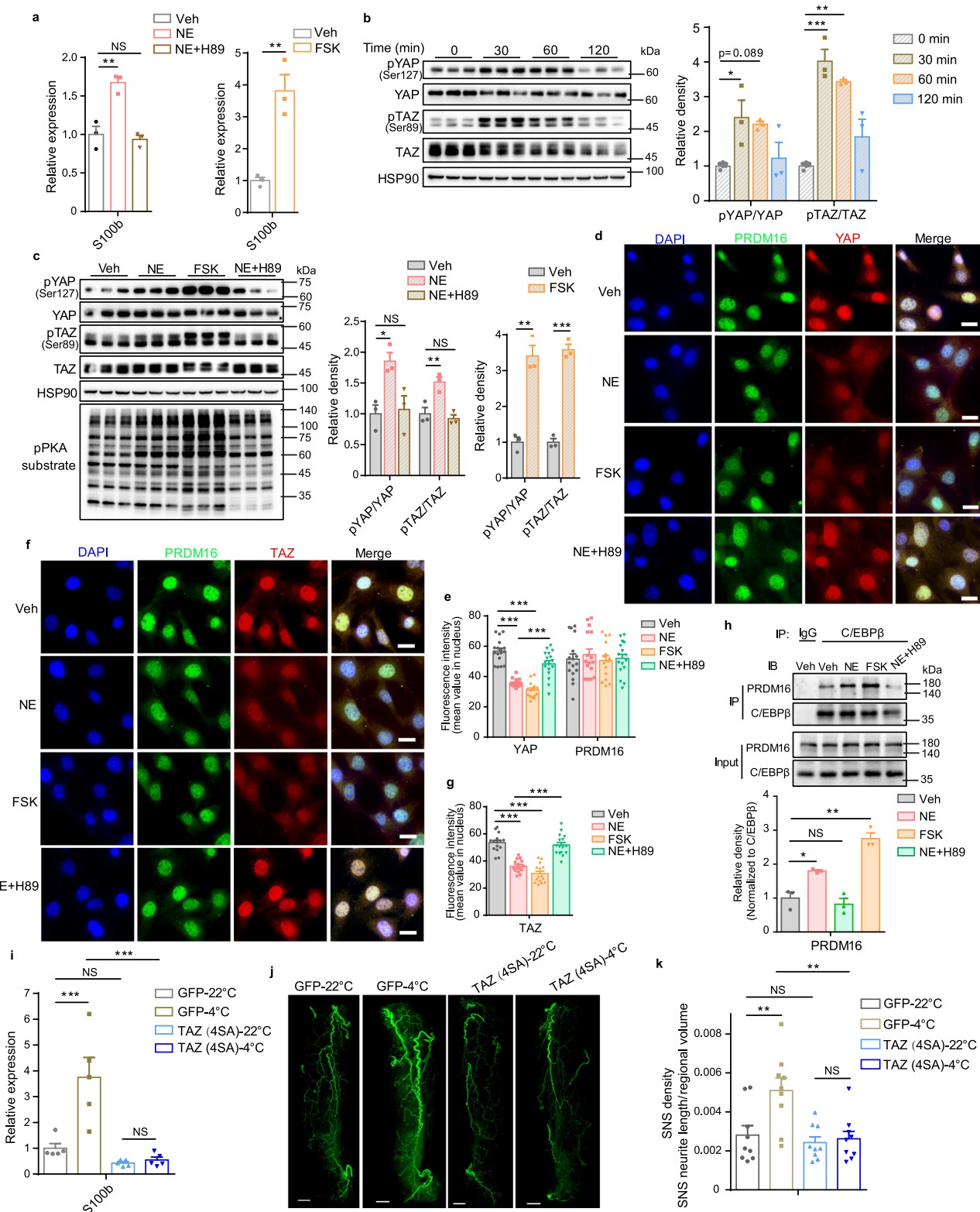

largely due to their less fat mass accumulation (Supplementary Fig. 8b). Accordingly, scWAT and eWAT had much lighter tissue weight and smaller adipocyte size, and YT-AKO mice had no fatty liver (Supplementary Fig. 8c, d). In contrast, YT-AKO mice had higher whole-body energy expenditure (VO2) but no change in food intake and locomotor activity (Supplementary Fig. 8e–g). Thermogenic genes like *Ucp1*, *Dio2*, and *Cox8b* were robustly upregulated, while pro-inflammatory genes including *Tnf*, *Il1b*, and *Nos2* were

largely downregulated in the scWAT of YT-AKO mice (Supplementary Fig. 8h, i). Besides, *S100b* in scWAT was upregulated and sympathetic innervation was also increased in YT-AKO mice (Supplementary Fig. 8h–k). Compared to control mice, there were no more lipids accumulated in sera of YT-AKO mice under HFD condition (Supplementary Fig. 8l, m). These beneficial alterations together markedly improved insulin sensitivity and glucose tolerance of YT-AKO mice (Supplementary Fig. 8n, o).

**Fig. 5 | NE-cAMP-PKA signaling stimulates YAP/TAZ phosphorylation to release their inhibition on S100b expression and sympathetic innervation. a** Relative mRNA level of *S100b* after treatment with indicated reagents for 12 h. **b** Time course of YAP and TAZ phosphorylation in response to NE treatment. **c** Phosphorylation of YAP and TAZ after treatment with NE, NE + H89, and forskolin (FSK) for 30 min, respectively. **d–g** Immunofluorescence staining of endogenous YAP (**d**), TAZ (**f**) and PRDM16 (**d, f**), and their quantification (**e, g**) after treated with indicated reagents for 30 min ($n = 17, 16, 15, 17$ cells for Veh, NE, FSK, NE + H89 group, respectively, in (**e**); $n = 15, 15, 15, 16$ cells for Veh, NE, FSK, NE + H89 group, respectively, in **g**). Scale bars: 20 μm. **h** Co-IP of endogenous PRDM16 and C/EBPβ and quantification after treated with indicated reagents for 30 min. **i** Relative mRNA level of *S100b* in scWAT injected with AAV-ADP-TAZ(4SA) or AAV-ADP-GFP for 3 weeks then under 22 °C and 4 °C for 2 days ($n = 5$ mice per group). **j, k** Whole-mount TH immunostaining (**j**) and quantification (**k**) of cleared scWAT as in (**i**) under 22 °C and 4 °C for 4 days ($n = 9$ random subregions from 3 independent tissues per group). Scale bars: 1 mm. Experiments in (**a–h**) were performed with SVF-derived beige adipocytes. $N = 3$ (**a–c, h**) biologically independent cell cultures per group, and data are mean ± s.e.m. Two-tailed unpaired Student's *t*-test [Veh versus FSK groups in (**a**), (**c**), (**e**), (**g**) and (**h**)]; one-way ANOVA with Tukey's multiple comparisons test (**a–c, e, g, h**); two-way ANOVA with Tukey's multiple comparisons test (**i, j**). *$P < 0.05$, **$P < 0.01$, ***$P < 0.001$; NS, not significant. Specific *p*-values and source data are provided as a Source data file.

Having established the prevention role of *Yap/Taz* loss in HFD-induced obesity, we next investigated whether their loss can reverse pre-established obesity. To this end, we used YT-iAKO mice fed HFD for 11 weeks to establish obesity, and then administrated with tamoxifen to delete *Yap/Taz* in adipocytes. Indeed, the pre-established obesity was reversed by 4 weeks of tamoxifen-induced *Yap/Taz* loss to a comparable level as control mice fed ND (Supplementary Fig. 9a). Similar to YT-AKO mice, YT-iAKO scWAT and eWAT had dramatically reduced tissue weight and smaller adipocyte size, and YT-iAKO mice had no fatty liver as well (Supplementary Fig. 9b, c), with slightly increased lipolytic ability in scWAT and eWAT (Supplementary Fig. 9d–g). YT-iAKO mice also had higher whole-body energy expenditure (VO2) but no change in food intake and locomotor activity (Supplementary Fig. 9h–j). Consistently, the expression of thermogenic genes and sympathetic innervation were robustly increased while inflammatory genes were largely downregulated in the scWAT of YT-iAKO mice (Supplementary Fig. 9k–m). Also, there was no lipid metabolite accumulation in sera of YT-iAKO mice under HFD feeding condition compared to those of controls (Supplementary Fig. 9n, o).

Adipose tissue inflammation is tightly associated with metabolic function. To dissect the inflammatory response of scWAT from YT-AKO and control mice, we analyzed the RNA-seq data of scWAT from adult YT-AKO and control mice fed normal diet. GSEA of acute and chronic inflammatory responses did not show significant difference between two groups, indicating that YAP/TAZ deletion does not influence inflammatory response but promote browning of scWAT under normal diet feeding condition (Supplementary Fig. 10a, b). On the other hand, WAT is infiltrated with pro-inflammatory M1 macrophages under HFD-induced or aging associated obesity condition[44,45]. As expected, we found that macrophages particularly M1 macrophages were less infiltrated into both scWAT and eWAT of YT-iAKO mice under HFD feeding condition. However, M1 macrophages were less infiltrated into only eWAT, the main expanded fat pad during aging (Supplementary Fig. 10c–k). In addition, percentage of M2 adipose tissue macrophages was decreased in old mice[45]. Consistently, percentages of M2 macrophages were significantly increased in scWAT and eWAT from YT-AKO aging mice, indicating an anti-inflammatory microenvironment in YT-AKO adipose tissues (Supplementary Fig. 10d–g).

**Loss of *Yap/Taz* promotes beige fat biogenesis in browning-reluctant visceral WAT**

Metabolic diseases come along with excessive accumulation of vWAT in obesity, including insulin resistance, diabetes and hyperlipidemia[46]. Although much less beige fat is induced upon adrenergic stimulation in vWAT than scWAT, it can profoundly ameliorate obesity and related metabolic dysfunction[8,47]. Through examining protein level of YAP/TAZ in different fat depots, we found that YAP/TAZ especially TAZ were greatly enriched in eWAT compared to BAT (Fig. 7a, b). Conversely, the phosphorylated and inactive YAP and TAZ were dramatically decreased in eWAT (Fig. 7a, c). There data indicated that active and stable YAP/TAZ were greatly accumulated in eWAT compared to BAT and scWAT. Since *Yap/Taz* loss induces beige fat biogenesis in scWAT, we then examined whether it also happens in eWAT. We found

that *Yap/Taz* loss profoundly induced beige fat formation in eWAT as shown by robustly increased expression of thermogenic genes like *Ucp1*, *Cox8b*, *elovl3* and *Dio2* after cold exposure for 1 week, with less dramatic change observed under room temperature condition (Fig. 7d, e and Supplementary Fig. 11a). Histologically, YT-AKO eWAT showed an evident beige fat morphology under cold condition compared to control mice (Fig. 7f). Whole-mount imaging of UCP1 confirmed much more beige fat biogenesis in the eWAT of YT-AKO mice (Fig. 7i and supplementary Movie 3). Mechanistically, *S100b* was dramatically induced in eWAT of YT-AKO mice under cold condition and the sympathetic innervation was greatly increased (Fig. 7d, g, h and Supplementary Fig. 11b), echoing previous reports showing that eWAT expresses less S100B and has less sympathetic innervation compared to scWAT[6,23].

Interestingly, *Yap/Taz* loss in adipocytes also resulted in profound beige fat biogenesis in eWAT of both age-associated and HFD-induced obese mice, while expression of inflammatory factors markedly decreased in HFD-induced obese mice (Supplementary Fig. 11c, d). Strikingly, eWAT had comparable expression of UCP1 as scWAT in YT-AKO mice fed HFD and in old mice (Supplementary Fig. 11e–h), indicating eWAT partially gained metabolically beneficial properties of scWAT upon *Yap/Taz* loss. Collectively, these findings suggest that YAP/TAZ deficiency can functionally turn inert and metabolically unhealthy visceral WAT into relatively active and metabolically benign even healthy subcutaneous WAT, which normalizes metabolic abnormalities under diet-induced obesity and aging conditions.

**AAV-*ADP-S100b* administration in visceral WAT relieves obesity and improves glucose homeostasis**

By reanalyzing the RNA-seq data of WAT from individuals with lean or obese phenotype (GSE141432), we found that S100B was greatly decreased while TAZ was increased in obese ones (Fig. 8a–c). But there was no change of PRDM16 and C/EBPβ (Fig. 8d, e). Furthermore, RNA-seq data in the GTEx database of visceral fat from young and old individuals revealed that S100B was downregulated while YAP/TAZ were upregulated in old ones (Supplementary Fig. 12a–c). Also, there was no change of PRDM16, accompanying with a little decrease in C/EBPβ (Supplementary Fig. 12d, e). These findings suggest that YAP/TAZ-S100B axis may serve as a therapeutic target in humans to activate inert and metabolically unhealthy visceral WAT and improve glucose homeostasis under diet-induced obesity and aging conditions.

To test the therapeutic potential, HFD-induced obese mice and old mice were administrated with AAV-*ADP-S100b* in eWAT, respectively (Fig. 8f and Supplementary Fig. 12f). GTT and ITT results reveal that mice from treatment group exhibited better glucose homeostasis compared to control group (Fig. 8g, h and Supplementary Fig. 12g, h). Correspondingly, the treatment group gained less body weight and visceral fat, which showed denser sympathetic innervation and higher expression of UCP1 (Fig. 8i–m and Supplementary Fig. 12i–m). We examined serum level of S100B and found that the upregulated S100B expression in eWAT did not affect circulating S100B level (Supplementary Fig. 12n), suggesting that S100B functions in WAT probably through a paracrine but not endocrine effect. All together, these data

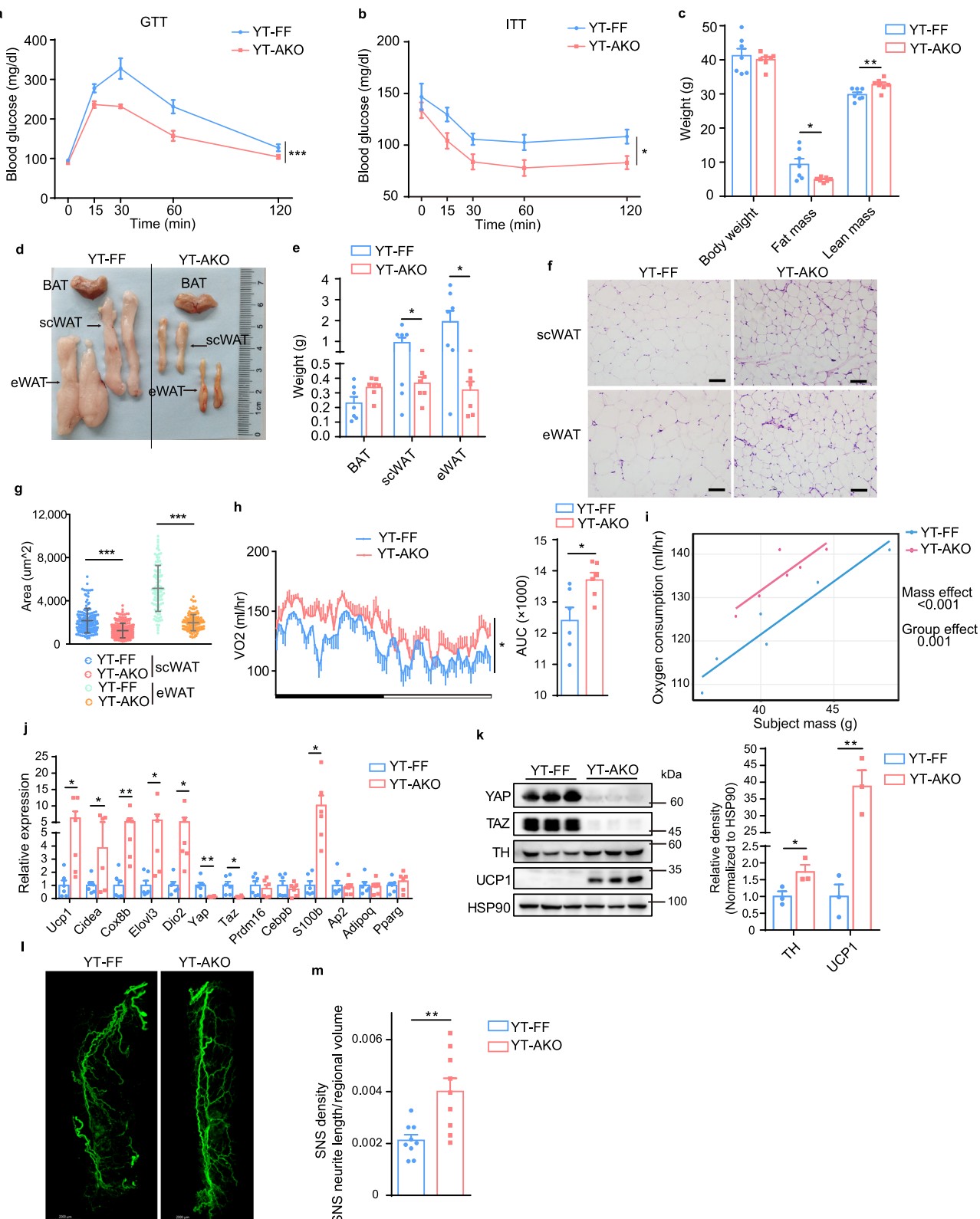

**Fig. 6 | Adipocyte-specific loss of *Yap/Taz* counteracts age-associated obesity and insulin resistance. a**, **b** Glucose tolerance test (GTT) (**a**) and insulin tolerance test (ITT) (**b**). **c** Body weight, fat mass and lean mass. **d**, **e** Gross view (**d**) and weight (**e**) of three adipose tissues. **f**, **g** H&E sections of scWAT and eWAT (**f**) and cell area quantification (**g**) (*n* = 101 cells in eWAT from YT-FF, 119 cells in eWAT from YT-AKO, 149 cell in scWAT from YT-FF and 204 cells in scWAT from YT-AKO group of three independent sections). Scale bars: 50 μm. **h**, **i** Oxygen consumption. **j** Relative mRNA levels of indicated genes in scWAT. **k** Immunoblot of indicated proteins in

scWAT. **l**, **m** Whole-mount TH immunostaining (**l**) and quantification (**m**) of cleared scWAT (*n* = 9 random subregions from 3 independent tissues per group). Scale bars: 2 mm. All the experiments were performed with 12-month-old YT-AKO and YT-FF male mice. *N* = 3 (**k**), 6 (**h**–**j**), 7 (**a**–**c**, **e**) mice per group, and data are mean ± s.e.m. Two-tailed unpaired Student's *t*-test (**c**, **e**, **g**, **h**−AUC, **j**, **k**, **m**); two-way repeated-measures ANOVA (**a**, **b**, **h**); CalR-ANCOVA (**i**). *$P$ < 0.05, **$P$ < 0.01, ***$P$ < 0.001. Specific $p$-values and source data are provided as a Source data file.

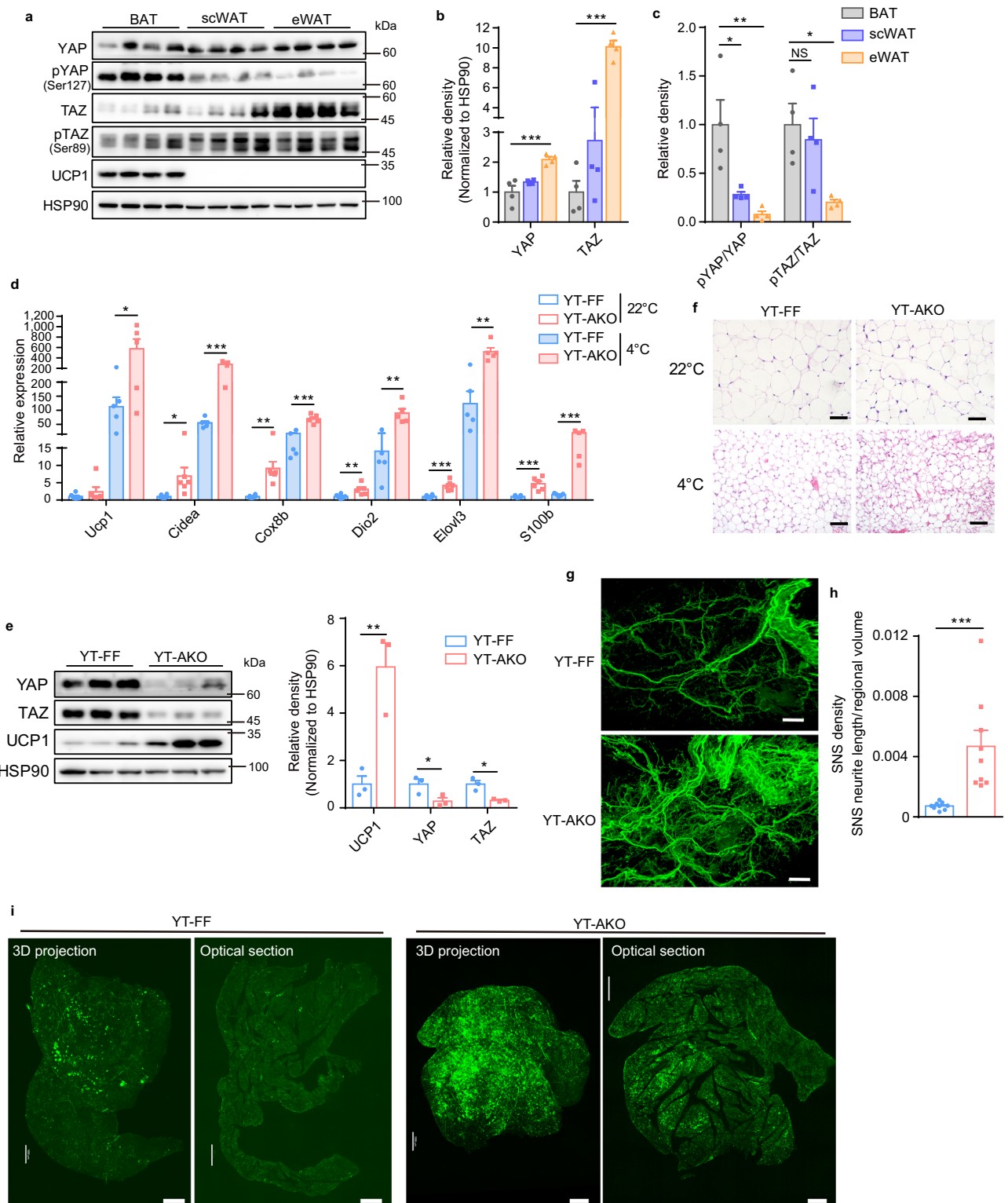

**Fig. 7 | *Yap/Taz* loss induces beige fat biogenesis in eWAT. a–c** Immunoblot (**a**) and quantification (**b**, **c**) of indicated proteins in three adipose tissues. **d** Relative mRNA levels of indicated genes in eWAT from mice under 22 °C or 4 °C for 7 days. **e** Immunoblot and quantification of indicated proteins in eWAT from mice post cold exposure for 7 days. **f** H&E sections of eWAT as in (**d**). Scale bars: 100 μm. **g**, **h** TH immunostaining (**g**) and quantification (**h**) of cleared eWAT by fluorescence light-sheet microscopy imaging (*n* = 9 random subregions from 3 independent

tissues per group). Scale bars: 0.5 mm. **i** UCP1 immunostaining of cleared eWAT as in (**e**) by fluorescence light-sheet microscopy imaging. Scale bars: 1 mm. *N* = 3 (**e**), 4 (**a**–**c**), 5 (**d**–4 °C), or 6 (**d**–22 °C) mice per group, and data are mean ± s.e.m. Two-tailed unpaired Student's *t*-test (**d**, **e**, **h**); one-way ANOVA with Bonferroni's multiple comparisons test (**b**, **c**). **P* < 0.05, ***P* < 0.01, ****P* < 0.001; NS, not significant. Specific *p*-values and source data are provided as a Source data file.

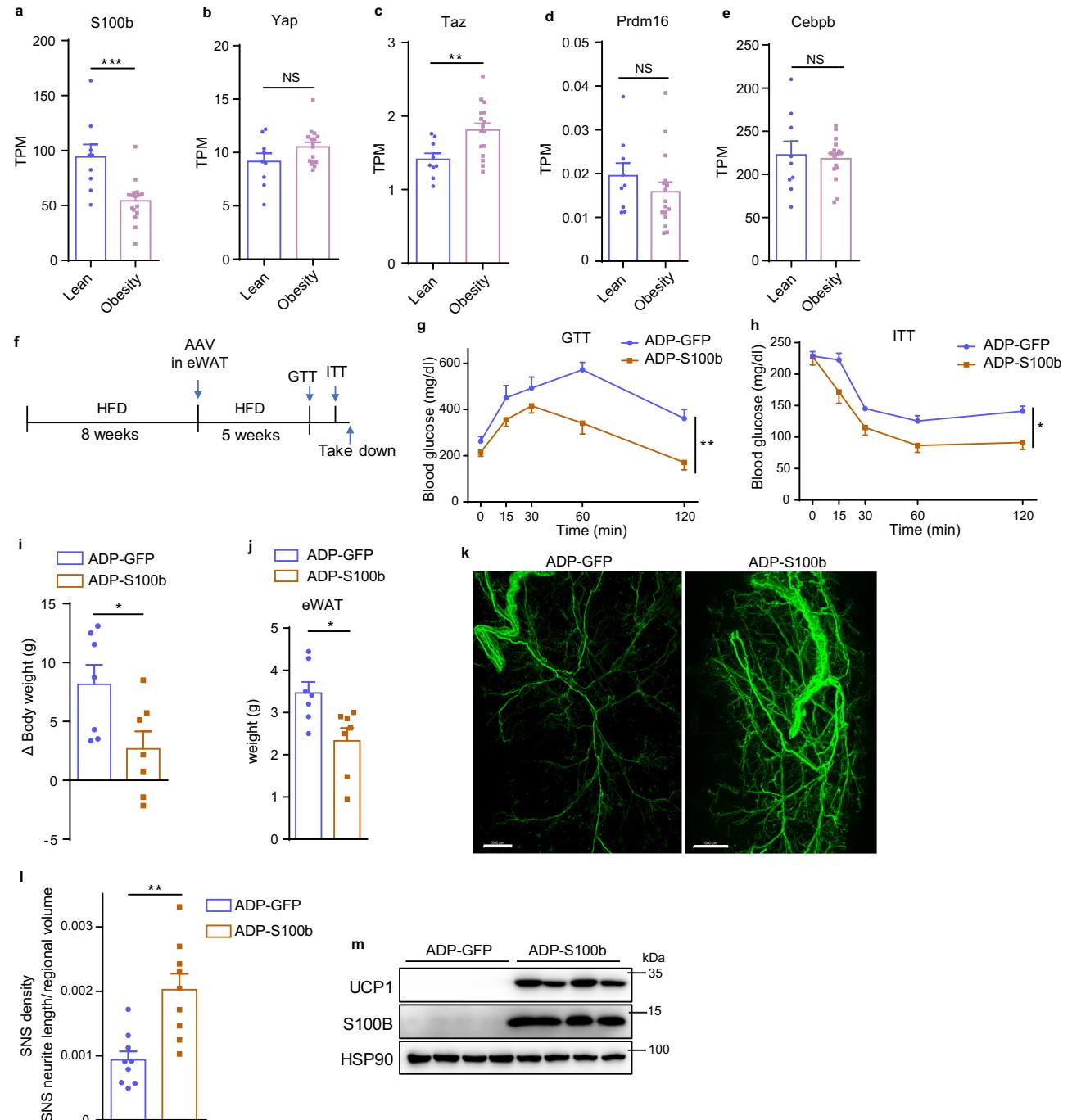

**Fig. 8 | AAV-*ADP-S100b* administration in visceral WAT restrains diet-induced obesity and improves glucose homeostasis. a–e** mRNA abundance (TPM) of indicated genes in WAT from individuals with lean or obese phenotype, measured by RNA-seq from GEO (GSE141432) (*n* = 9 individuals in lean group and 16 individuals in obesity group). **f** Schematic diagram of AAV administration in eWAT from HFD-induced obese mice. **g, h** Glucose tolerance test (GTT) (**g**) and insulin tolerance test (ITT) (**h**) of mice as in (**f**). **i, j** Weight gain of HFD-induced obese mice (**i**)

and tissue weight of eWAT (**j**) upon AAV administration. **k, l** Whole-mount TH immunostaining (**k**) and quantification (**l**) of cleared eWAT upon AAV administration (*n* = 9 random subregions from 3 independent tissues per group). Scale bars: 1 mm. **m** Immunoblot of indicated proteins in eWAT upon AAV administration. *N* = 4 (**m**) or 7 (**g–j**) mice per group, and data are mean ± s.e.m. Two-tailed unpaired Student's *t*-test (**a–e**, **i**, **j**, **l**); Two-way ANOVA (**g**, **h**). *$P < 0.05$, **$P < 0.01$, ***$P < 0.001$. Specific *p*-values and source data are provided as a Source data file.

indicate that ectopically expressing S100B in vWAT by AAV could be an effective and safe means to treat obesity and associated metabolic disorders.

## Discussion

PRDM16 plays a dominant role in brown and beige fat development and function through transcriptional regulation of their cell fate determination, differentiation, and identity maintenance. PRDM16 acts

as a coactivator by collaborating with master transcriptional factors like PPARγ and PGC1α to broadly elicit thermogenic program in brown and beige progenitors and mature cells[11]. MicroRNA-mediated transcription and E3 ligase-mediated protein degradation has been established to regulate PRDM16 level[48,49]. Here, we identified that YAP/TAZ acts as a molecular switch of beige fat innervation through their association and dissociation with transcriptional coactivator PRDM16, which controls PRDM16-C/EBPβ complex formation to regulate *S100b*

expression and sympathetic innervation. Previous study reported that TLE3 disrupts PRDM16-PPARγ complex to broadly and directly suppress thermogenic program in a beige adipocyte-autonomous way[50]. In our study, YAP/TAZ blockade of PRDM16-C/EBPβ complex formation did not directly affect thermogenic program in beige adipocytes, instead it specifically inhibited *S100b* expression to impair sympathetic innervation and thus indirectly suppressed beige fat biogenesis and thermogenesis.

Sympathetic nerves play critical roles in lipolysis and beige fat biogenesis upon cold exposure and hormone stimulation[5]. Selective sympathetic denervation in scWAT and BAT impairs cold-induced thermogenesis[24,51]. As a browning-resistant fat, eWAT has much less sympathetic nerves than scWAT, making it rational to increase eWAT innervation for enhanced beige fat biogenesis and energy expenditure[23]. Previous studies established that cold exposure triggers sympathetic expansion in WAT, but the underlying mechanisms are poorly understood[24,52]. PRDM16 promotes neurotrophic factor S100B expression and sympathetic innervation of WAT also through unknown mechanisms[6,23]. Here, we revealed that cold-elicited cAMP-PKA signaling phosphorylates and inactivates YAP/TAZ, which impedes their binding to PRDM16 and thus enhance PRDM16-C/EBPβ complex formation for *S100b* expression and sympathetic innervation in adipose tissue (Supplementary Fig. 13).

In our study, we did not observe any induction of thermogenic genes in BAT from YT-AKO mice compared to controls, probably due to two reasons. First, BAT has much more PRDM16 but less non-phosphorylated and active YAP/TAZ than scWAT and eWAT[8], thus YAP/TAZ cannot compete with overwhelming amount of PRDM16 in BAT for C/EBPβ association, and that YAP/TAZ deletion in BAT did not influence PRDM16-C/EBPβ interaction (Supplementary Fig. 4m). Second, PRDM16 deletion in adipocytes displays a relatively normal pattern of gene expression in BAT, suggesting other compensatory mechanisms are involved in BAT[9]. Tharp et al. used *Ucp1^Cre* to delete one copy of *Yap/Taz* in brown fat, and they proposed that YAP/TAZ can promote *Ucp1* expression and thermogenic capacity in brown adipocytes in a TEADs-dependent way[53]. However, we used the same *Ucp1^Cre* to even delete two copies of *Yap/Taz* and observed no alteration of *Ucp1* expression except a neurological symptom (Supplementary Fig. 1e, and supplementary Movie 1), which was probably attributed to a leak of *Ucp1^Cre* expression in the central nervous system that affected the development of brain like cerebellum and whole-body metabolism including thermogenesis[54-56]. Wang et al. reported that YAP/TAZ can protect against white adipocyte death and YAP/TAZ loss in adipocytes induces cell apoptosis and impedes fat mass accumulation under HFD feeding condition[15]. However, they did not explain where to go and how to be utilized of released lipids from dead cells. Since lipolytic ability of WAT from YT-AKO did not enhance markedly, our data suggest that the released lipids from apoptotic adipocytes are utilized as fuels for increased beige fat.

By adipocyte-specific deletion of YAP/TAZ, we prove in concept that inhibiting YAP/TAZ in adipose tissues particularly in visceral white adipose tissue could be an effective therapeutic approach to treat diet-induced or aging-related obesity and their associated metabolic disorders. Furthermore, Hippo signaling inactivation is associated with various cancers and high nucleus levels of YAP/TAZ expression correlate with poor prognosis and therapeutic resistance[57]. Thus, targeting the Hippo pathway particularly YAP/TAZ could not only improve metabolism abnormalities but also enhance therapeutic effects in some aggressive cancers. RNA interference (RNAi) has been demonstrated to be a powerful therapeutic strategy and the advancements in pharmaceutical chemistry bring RNAi-based drugs into clinical application[58]. Besides, proteolysis targeting chimeras (PROTACs) technology is an emerging technology, which can effectively promote degradation of proteins that are difficult to target with small molecules[59]. All these technologies provide promising means to target YAP/TAZ for the prevention and treatment of obesity and associated metabolic disorders and cancers. In this study, we targeted S100B-the downstream of YAP/TAZ by adipocyte-specific promoter-driven expression in eWAT, which improved obesity and associated metabolic disorders.

In summary, our study revealed that YAP/TAZ interact with PRDM16 to disrupt PRDM16-C/EBPβ complex, which represses *S100b* expression and sympathetic innervation expansion. Enhancement of sympathetic innervation through targeting the YAP/TAZ-S100B axis promotes beige fat biogenesis and energy expenditure, which ameliorate diet-induced or age-associated obesity and their concomitant metabolic dysfunction.

## Methods

### Mice

Animal studies were in compliance with the regulations by the Institutional Animal Care and Use Committee (IACUC) of Peking University (IMM-QiuYF-1). All mice were housed at temperature ($22 \pm 1\,°C$) and humidity ($60\% \pm 10\%$) controlled environment under 12 h–12 h light-dark cycle, and had free access to water and normal chow diet (Xietong Shengwu, 1010063). Unless otherwise specified, 8–10-week-old male mice were used in animal experiments. *Yap1^{fl/fl}* (027929, C57BL/6–129), *Adipoq^{Cre}* (028020, C57BL/6), *Ucp1^{Cre}* (024670, C57BL/6), *Adipoq^{CreERT2}* (025124, C57BL/6), *E2a^{Cre}* (003724, C57BL/6), and Rosa26-LSL-Cas9 knock-in (026175, C57BL/6) mice were purchased from The Jackson Laboratories. *Taz^{fl/fl}* mice (C57BL/6–129) were kindly provided by Nan Tang (National Institute of Biological Sciences, Beijing, China). YT-AKO, YT-UKO and YT-iAKO mice were generated by crossing *Yap1^{fl/fl};Taz^{fl/fl}* mice with *Adipoq^{Cre}*, *Ucp1^{Cre}*, *Adipoq^{CreERT2}* mice, respectively. *Rosa26-LSL-Cas9;Adipoq^{Cre}* and *Cas9^{Tg/Tg}* mice were generated by crossing Rosa26-LSL-Cas9 knock-in mice with *Adipoq^{Cre}* and *E2a^{Cre}* mice, respectively.

For conditional knockout of *Yap* and *Taz*, 100 mg/kg tamoxifen (20 mg/mL in corn oil) (Sigma-Aldrich) was i.p. injected into 8-week-old male mice for 3 doses every other day, tissues were collected after 4 weeks. In HFD-induced obesity model, 100 mg/kg tamoxifen (20 mg/mL in corn oil) was i.p. injected for 5 doses every day into male YT-iAKO and YT-FF mice fed with HFD (Research Diets, D12492) for 11 weeks.

Unless specified, animals used in the experiment are male mice. Mice were euthanized via asphyxiation using CO2 followed by cervical dislocation.

### Plasmids construction

*Yap*, *Taz*, *Yap(S79A)*, *Taz(S51A)*, *Yap(5SA)*, *Taz(4SA)*, *Cebpa*, *Cebpb*, *Pparg*, *S100b* and *Prdm16* cDNAs were cloned into pAAV-CAG-MCS vector, which was modified from pAAV-CAG-GFP (Addgene, 37825), to generate pAAV-CAG-Yap, pAAV-CAG-Taz, pAAV-CAG-Yap(S79A), pAAV-CAG-Taz(S51A), pAAV-CAG-Yap(5SA), pAAV-CAG-Taz(4SA), pAAV-CAG-Cebpa, pAAV-CAG-Cebpb, pAAV-CAG-Pparg, pAAV-CAG-S100b and pAAV-CAG-Prdm16. Cyclization recombinase (*Cre*), *Taz(4SA)*, *Cebpb*, and *S100b* cDNAs were cloned into pAAV-ADP-MCS vector[16] to generate pAAV-ADP-Cre, pAAV-ADP-Taz (4SA), pAAV-ADP-Cebpb, pAAV-ADP-S100b. As for AAV-mediated guide RNA plasmid construction, the indicated 20 bp gRNA was inserted into pAAV-U6-sgRNA-CMV-GFP vector (Addgene, 85451). To simultaneously knockout multiple genes, gRNAs targeting different genes were linked by 20 bp linkers. gRNAs sequences used in this study are listed in supplementary Table 1.

### Differentiation of primary preadipocytes

For the differentiation of primary preadipocytes, pooled scWAT was isolated from P14-P21 WT mice and digested by Collagenase Type I

(2 mg mL$^{-1}$, Sigma-Aldrich) in SVF buffer (1.1 mM CaCl$_2$, 2.7 mM KCl, 118 mM NaCl, 0.5 mM MgCl$_2$, 0.4 mM NaH$_2$PO$_4$, 20 mM HEPES pH 7.2, 5.5 mM glucose and 1% BSA (fatty-acid free)) for 50 min at 37 °C, shaking at 120 r.p.m. Stromal vascular fraction (SVF) suspension was filtered through a 75-μm cell strainer and then centrifuged to collect pellet, which was then resuspended and plated in DMEM supplemented with 10% FBS, 20 mM HEPES pH 7.2 and 1% P/S at 37 °C in a humidified 5% CO$_2$ incubator. When reaching confluence, the cell culture was replaced by Induction Medium (5 μg mL$^{-1}$ insulin, 1 nM T3, 125 μM indomethacin, 1 μg mL$^{-1}$ dexamethosone, 0.5 mM 3-isobutyl-1-methylxantine and 0.5 μM rosiglitazore). After 48 h, the culture was changed to Maintenance Medium (5 μg mL$^{-1}$ insulin, 1 nM T3, and 0.5 μM rosiglitazore) and cultured for another 4–5 days followed by other beige cell experiments.

## RNA isolation and RT-qPCR
Total RNA was isolated from cells or tissues using TRIzol (Sigma-Aldrich) according to the manufacturer's instruction. For reverse transcription, 1 ug RNA was used to generate cDNA using the 5×All-In-One MasterMix kit (abm) and RT-qPCR was carried out with ChamQ SYBR qPCR master mix (Vazyme) on ABI StepOnePlus (Applied Biosystems). Relative expression level of mRNAs was calculated by the 2$^{(-\Delta\Delta CT)}$ method and 36B4 (Rplp0) was used as an internal control. Primer sequences are listed in Supplementary Table 2.

## CUT&Tag and RT-qPCR
CUT&Tag was performed by using Hyperactive Universal CUT&Tag Assay Kit (Vazyme, TD903) according to the manufacturer's protocols. In brief, in the 6th day of differentiation, the SVF-derived beige adipocytes were digested from the cell plate, and around 5 × 10$^5$ cells in 1.5 mL tude per group were used to do the assay. After binding to ConA beads, the nuclei of adipocytes were incubated at 4 °C overnight with anti-PRDM16 (R&D, 1:50) or anti-C/EBPβ (Santa Cruz Biotechnology, 1:100) and the corresponding IgG control, respectively. Washing for 3 times, the nucleus-ConA beads were incubated with anti-sheep or anti-mouse secondary antibody for 1 h at 22 °C. After washing, the nucleus-ConA beads was incubated with pA/G-Tnp and get the target DNA fragments followed by DNA extraction procedure.

For DNA library amplification, the extracted DNA was used to do PCR for 15 cycles with adaptor primers as F: TCGTCGGCAGCGTCA GATGTGTATAAGAGACAG, R: GTCTCGTGGGCTCGGAGATGTGTAT AAGAGACAG, followed by DNA product purification procedure. RT-qPCR was performed to detect the relative enrichment of targeted DNA product. The qPCR primers used for the indicated promoter region of *S100b* are F: 5'-CACTTCCTACCCAACAGACC-3', R: 5'-AGGAT AGGGAGGAGTTAGACAC-3'.

## Oil red O staining
SVF-derived beige adipocytes were washed twice with PBS and then fixed with 4% paraformaldehyde (PFA) for 30 min at 22 °C. After washing with PBS, adipocytes were incubated with Oil red O working solution (0.5 mg/ml Oil red O) for 30 min and washed with 75% ethanol 3 times (1 min per time), then visualized by ordinary optical microscopy. For quantification, the stained adipocytes were incubated with isopropanol for 10 min, and the absorption value of solution was measured by spectrophotometer at 540 nm.

## Mouse S100B ELISA
Mouse serum S100B ELISA was performed as per the guidance from manufacturer. Briefly, mouse blood was collected with heparin-coated tube and then centrifuged to get serum. Experimental and standard samples were added to anti-S100B antibody-coated plate and then added HRP-anti-S100B antibody, incubating for 60 min at 37 °C. Plate was washed for 5 times and added with substrates, incubating for

15 min at 37 °C. Finally, stop buffer was added to plate, which was then detected by microplate reader.

## Isolation of SVF and adipocytes
scWAT from indicated mice were minced and digested with Collagenase Type I (180 U/mL, Worthington) in SVF buffer at 37 °C with agitation for 30 min, 150 r.p.m. Cell suspension was filtered through a 75-μm strainer and then centrifuged at 520 × $g$ for 5 min. The floated cells were collected as adipocytes and the pelleted cells were resuspended in red blood cell lysis buffer (0.15 M NH4Cl, 10 mM NaHCO$_3$, 1.1 mM EDTA) for 5 min. After centrifugation at 520 × $g$ for 5 min, the pellet was collected as SVF.

## HEK-293T cell culture
HEK-293T (ATCC, CRL-3216) was cultured in DMEM added with 10% FBS and 1% penicillin/streptomycin. Cells were maintained at 37 °C in a humidified atmosphere with 5% CO$_2$.

## Immunoblot and immunoprecipitation
Tissue was homogenized in high-salty RIPA buffer (450 mM NaCl, 1% NP-40, 0.1% SDS, 0.5% Deoxycholic acid (sodium salt), 50 mM Tris-HCl pH 7.5, and cocktail protease inhibitors) and cells were lysed in low-salty RIPA (150 mM NaCl, 1% NP-40, 0.1% SDS, 0.5% Deoxycholic acid (sodium salt), 50 mM Tris-HCl pH 7.5, and cocktail protease inhibitors). Protein concentration was measured by BCA method (Cwbiotech). Total protein was separated by SDS-PAGE and transferred to nitrocellulose membrane (Merck). Membranes were incubated with primary antibodies overnight at 4 °C and then with HRP-conjugated anti-rabbit IgG, anti-mouse IgG (Thermo Fisher) or anti-sheep IgG (Merck Millipore) secondary antibody for 1 h at 22 °C. The blot signal was detected by the automatic chemiluminescence analysis system (Tanon) with Super ECL Detection Reagent (Yeasen). Cytoplasmic and nuclear fractionation was performed with Nuclear and Cytoplasmic Protein Extraction Kit (Beyotime) according to the manufacturer's protocol. All quantifications of protein levels were performed with ImageJ software.

For immunoprecipitation experiments, cells were lysed with immunoprecipitation lysis buffer (120 mM NaCl, 50 mM Tris-HCl pH 8.0, 10% glycerol, 1% NP-40, 5 mM EDTA, and cocktail protease inhibitors) for 1–2 h at 4 °C and then centrifuged at 4 °C for 10 min. For HA-tagged overexpression cell lysis, the supernatant was incubated with anti-HA Magnetic Beads (Thermo Fisher Science, 88836) for 2 h at 4 °C. For endogenous-protein immunoprecipitation, the supernatant was incubated with indicated primary antibodies at 4 °C overnight and then added with Protein G Beads (Cell Signaling Technology, 37478) for 3 h at 4 °C. The beads were washed with immunoprecipitation lysis buffer for three times and the bound protein was analyzed by immunoblot.

The following antibodies and dilutions were used: anti-UCP1 (Sigma-Aldrich, U6382, 1:3000 for scWAT and cells, 1:20,000 for BAT), anti-PRDM16 (R&D, AF6295, 1:400), anti-C/EBPβ (Santa Cruz Biotechnology, sc-7962x, 1:10,000), anti-Perilipin (Vala Science, 4854, 1:10,000), anti-phospho-Perilipin (Ser 522, Vala Science, 4856, 1:10,000), anti-Lamin B1 (Protein Tech, 66095-1-Ig, 1:10,000), anti-α-tubulin (Sigma-Aldrich, T6199, 1:10,000), anti-TH (Millipore, AB1542, 1:3000), anti-HSP90α/β (Santa Cruz Biotechnology, sc-13119, 1:10,000), anti-Flag (Abmart, 293881, 1:5000), goat anti-mouse IgG:HRP (Thermo Scientific, 32430, 1:10,000), goat anti-rabbit IgG:HRP (Thermo Scientific, 31460, 1:10,000) and donkey anti-sheep IgG:HRP (R&D,

HAF016, 1:500). The followings were from Cell Signaling Technology: anti-YAP (4912 or 14074, 1:1000), anti-TAZ (4883 or 83669, 1:1000), anti-phospho-YAP (Ser 112, 4911, 1:1000), anti-phospho-TAZ (Ser 89, 59971, 1:1000), anti-HSL (4107, 1:1000), anti-phospho-HSL (Ser 660, 4126, 1:1000), anti-S100B (9550, 1:1000), anti-C/EBPα (2295,

1:1000), anti-PPARγ (2435, 1:1000), anti-PKA substrate (RRXS*/T*, 9624, 1:1000), anti-HA (3724, 1:1000).

## Flow cytometry

SVF was isolated from scWAT or eWAT and resuspended in red-blood-cell lysis buffer for 5 min to remove blood red cells. The collected cell pellet was incubated with LIVE/DEAD™ Fixable Violet Dead Cell Stain (Thermo Scientific, L34955, 1:1000) at 22 °C for 10 min for live-dead staining. After washing with PBS, the cell pellet was blocked by anti-CD16/32 (BioLegend, 1:1000) and then incubated with the following antibodies for 20 min at 4 °C: anit-CD45-APC Cy7 (BioLegend, 103116, 1:300), anti-F4/80-FITC (Cell Signaling, 52267, 1:100), anti-CD11c-AF647 (BioLegend, 117314, 1:100), anti-CD206-PE Cy7 (BioLegend, 141720, 1:100). After washing and resuspended with FACS buffer, cells were subjected to analyze by CytoFLEX S flow cytometer (Beckman Coulter) and sequential plot analysis was performed with the Flowjo software (v10).

## AAV and lentivirus production

The pAAV was co-transfected into HEK-293T cells with Helper and RC2/8 or DJ plasmid to generate AAV for in vivo (RC2/8) and in vitro (DJ) experiments. Sixty hours post transfection, cells were collected and resuspended in AAV lysis buffer (150 mM NaCl, 20 mM Tris pH 8.0). Virus particles were released from cells through repeated freezing and thawing and purified via discontinuous iodixanol gradient centrifugation. The titer was determined by RT-qPCR.

For lentivirus production, the *Prdm16* and *Prdm16 (R998Q)* cDNAs were inserted into pPWI vector (Addgene, 12254), respectively, to generate pWPI-Prdm16-FLAG and pWPI-Prdm16 (R998Q)-FLAG. The pWPI was then co-transfected into HEK-293T with psPAX2 and pMD2.G. The supernatant was collected 48 h post transfection and purified via sucrose gradient centrifugation. The virus titer was determined by RT-qPCR.

## Mass spectrometry

To identify potential PRDM16-interacting proteins in beige adipocytes, primary preadipocytes from scWAT were infected with *Prdm16*-FLAG-lentivirus and then differentiated into beige adipocytes. In the 7th day during differentiation, adipocytes were lysed for immunoprecipitation with anti-FLAG agarose beads (Smart Lifesciences, SA042001). The immunoprecipitated was separated through SDS-PAGE. After Coomassie blue staining, bands of interest were cut from the gel and digested with sequencing grade-modified trypsin. The peptides were analyzed with liquid chromatography–tandem mass spectrometry (LC–MS/MS) using Orbitrap Q Exactive mass spectrometer at NCPS at Peking University. UniProt mouse database was used to search the LC–MS/MS data.

## Glucose homeostasis test in mice

For the HFD-induced obesity model, mice were fed a HFD started at the age of 6 weeks. For GTT, HFD-fed or ND-fed old and young mice were fasted for 8 h (from 7:00–15:00) and then i.p. administered with glucose (Sigma-Aldrich) (2 g kg$^{-1}$ body weight). For ITT, HFD-fed or ND-fed old and young mice were fasted for 6 h (from 9:00–15:00) and then i.p. administered with insulin (Eli Lilly) (0.7 U kg$^{-1}$ body weight for ND-fed young mice, 1 U kg$^{-1}$ body weight for ND-fed old mice and 1.5 U kg$^{-1}$ body weight for HFD-fed mice). Tail bleeds were used to measure blood glucose levels at indicated time points.

## Energy expenditure in mice

Whole-body energy expenditure (VO$_2$, VCO$_2$), respiration exchange rate (RER) and locomotor activity of ND-fed or HFD-fed old and young mice were monitored by Comprehensive Lab Animal Monitoring System (CLAMS, Columbus Instruments). Prior to measurement, mice

were acclimated for 24 h. The regression-based analyses of oxygen consumption in mice were performed by CaIR-ANCOVA (https://calrapp.org/)[60]. Fat mass and lean mass of mice were measured by Dual-energy X-ray absorptiometry (DXA).

## Body temperature measurement

Mice housed at 22 °C were transferred to 4 °C cages individually that had been pre-chilled, with free access to water and food. Rectal temperature was monitored at indicated time points using a rectal temperature probe (Physitemp). Once core body temperature dropped below 30 °C, mice would be euthanized.

## CL-316,243-induced energy expenditure

To measure CL-316,243-induced oxygen consumption, mice were transferred to individual cages of CLAMS at 30 °C 1 day before experiment to acclimate the environment, with free access to water and food. Next day, the mice were i.p. injected with CL-316,243 (1 mg kg$^{-1}$ body weight) and oxygen consumption was recorded.

## Histology and cell immunofluorescence

Tissues were fixed in 4% paraformaldehyde (PFA) for 24 h, and then dehydrated and embedded in paraffin. The paraffin blockers were cut into 5 μm sections and stained with hematoxylin and eosin. The H&E images were taken from histological light microscopy (Olympus BX51) and adipocyte diameter was measured by imageJ software.

For immunofluorescence staining, SVF-derived beige adipocytes were washed twice with PBS and then fixed with 4% paraformaldehyde (PFA) for 20 min at 22 °C. After washing, the fixed cells were permeabilized with 0.5% Triton X-100 (Lablead) for 20 min and then blocked with 5% BSA for 30 min at 22 °C. After washing, the adipocytes were incubated with anti-PRDM16 (R&D, 1:50), anti-YAP (CST, 14074, 1:100) or anti-TAZ (CST, 83669, 1:100) at 4 °C overnight. After washing with PSBT, the adipocytes were incubated with Alexa Fluor 488-conjugated anti-sheep (Invitrogen, A11015, 1:300) and Alexa Fluor 594-conjugated anti-rabbit (Abcam, ab150080, 1:300) for 1 h at 22 °C, followed by counterstaining with DAPI (CST, 1:5000). Adipocytes were imaged using Nikon A1RSi+ Confocal Imaging Systems for Fluorescence.

## Whole-mount immunostaining, tissue clearing, and imaging

The published detailed procedures had been followed in this study[23,24]. In brief, mice were anesthetized and perfused with 10 μg/mL heparin-contained PBS firstly. Then scWAT or eWAT was dissected and fixed with 4% PFA (Beyotime) at 4 °C overnight. After washing with PBS, the tissues were dehydrated with 20%, 40%, 60%, 80%, 100% methanol in B1N buffer (0.1% Triton X-100/0.3 M glycine, PH7.0) sequentially and delipidated with DCM (Sigma-Aldrich). After bleaching with 5% H$_2$O$_2$, the tissues were rehydrated with 80%, 60%, 40%, 20%, 0% methanol in B1N buffer successively and incubated with PTxwH buffer (0.1% Triton X-100/0.05% Tween 20/2 μg/mL heparin in PBS) for 2 h before staining. In staining step, the samples were incubated with anti-TH (Millipore, AB1542, 1:300) or anti-UCP1 (Abcam, ab10983, 1:100) for 3 days at 22 °C and then with Alexa Fluor 488-conjugated anti-sheep (Invitrogen, A11015, 1:400) or Alexa Fluor 488-conjugated anti-rabbit (Cell Signaling Technology, 4412, 1:400) for 3 days at 22 °C after washing with PTxwH buffer. Tissues were dehydrated with 25%, 50%, 75%, 100% methanol in H$_2$O sequentially after staining and then washed with 100% DCM, followed by cleaning with dibenzyl ether (DBE, Sigma-Aldrich) overnight. The volume imaging was performed in Zeiss Light Sheet Z.1, and Imaris software was used to process images. Notably, compared to stronger TH signal in scWAT, UCP1 signal is weaker and thus the background signal derived from autofluorescence especially in lymph node of scWAT is obvious.

## RNA-seq and analysis

Total RNA was isolated from scWAT of YT-FF and YT-AKO male mice (*n* = 3 per group) under 22 °C and subjected to commercial RNA-seq analyses (Frasergen Bioinformatics) with pair-end sequencing on Illumina NovaSeq 6000 instrument. Mouse genome (mm10) was used to map the seq reads with HISAT2 (v2.1.0, https://daehwankimlab.github.io/hisat2/). DEGs were generated by DEseq2 (V2.11.40.6, https://bioconductor.org/packages/release/bioc/html/DESeq2.html) and subjected to Kobas (http://kobas.cbi.pku.edu.cn/) pathway analysis website to get the pathway enrichment information. The heat map of thermogenesis pathway was performed by Hiplot (https://hiplot-academic.com/) and the Padj was calculated by the clusterProfiler R package (https://guangchuangyu.github.io/software/clusterProfiler/). The GSEA was performed by GSEA software (https://www.gsea-msigdb.org/gsea/index.jsp).

## Quantification and statistical analysis

The number of mice or samples per group, replicates of independent experiments and statistical tests are shown in the figure legends. The chosen of sample size was based on the pilot experiments or studies published[16,61] and to ensure statistical possibility, meanwhile to minimize animal usage in the experiments based on the 3R principles. Statistic analyses were performed by GraphPad Prism 8 or Microsoft Excel 2019. The statistical methods are indicated in the figure legend.

## Reporting summary

Further information on research design is available in the Nature Portfolio Reporting Summary linked to this article.

## Data availability

The RNA seq data generated in this study has been deposited in the Gene Expression Omnibus (GEO) under accession number GSE222069 and are publicly available as of the date of publication. The ChIP-seq data used in this study were available in GEO under accession number GSE63965 (PRDM16), GSM686970 (C/EBPβ), and GSM1218859 (C/EBPα). The RNA seq data of WAT from human were available in GEO under accession number GSE141432 and from GTEx database (https://www.gtexportal.org/home%5B33). The potential transcriptional factors binding to SP was predicted by PROMO (https://alggen.lsi.upc.es/cgi-bin/promo_v3/promo/promoinit.cgi?dirDB=TF_8.3/). Source data are provided with this paper.

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

## Acknowledgements

We thank members of the Qiu laboratory for comments on the manuscript. We thank National Center for Protein Sciences at Peking University for assistance with Nikon A1RSi+ Confocal Imaging, image processing and protein MS analysis. We also thank Dr. Wenwen Zeng and Core Facility of Center of Biomedical Analysis at Tsinghua University and Light Innovation Technology Ltd for assistance with Light Sheet Microscopy. This work was supported by grants from National Key R&D Program of Ministry of Science and Technology of the People's Republic of China (2021YFA0804801 and 2018YFA0800702), National Natural Science Foundation of China (91642113 and 31671227) to Y.Q.

## Author contributions

X.H. and D.W. designed and performed the main experiments with assistance from X.L., H.S., Y.Z., Z.Z., Y.W., J.Y., and K.X.; X.H., D.W., and Y.Q. discussed and interpreted the results from the study; X.H., D.W., and Y.Q. conceived, supervised, and wrote the paper.

## Competing interests

The authors declare no competing interests.
