## [Peer Review File · Nature Communications]

Transcriptional repression of beige fat innervation via a YAP/ TAZ-S100B axisEditorial Note: Parts of this Peer Review File have been redacted as indicated to remove third-party material where no permission to publish could be obtained.

REVIEWER COMMENTS

Reviewer #1 (Remarks to the Author):

In this manuscript, the authors investigated the role of YAP/TAZ in regulating the sympathetic innervation of adipose tissue via effects on S100b. They began by identifying YAP/TAZ as interacting with PRDM16 in adipocytes. Next, they made mice with adipocyte-specific deletion of YAP/TAZ and saw an increase in thermogenic beige fat, with associated metabolic benefits. Similar findings were reported with temporal deletion of YAP/TAZ in adipocytes using an AAV or a tamoxifen inducible system. Because in vitro differentiated adipocytes from YAP/TAZ knockout animals had no increase in thermogenic gene expression, the authors postulated that the in vivo phenotype was the result of a non-cell autonomous effect. So, they went on to investigate sympathetic innervation of adipose tissue. Their data suggests that YAP/TAZ deletion in adipocytes does result in increased SNS innervation, and profiling identified S100b as a possible mediator. Functional experiments were done to show that S100b can in fact promote sympathetic innervation. The authors then present a model, whereby PRDM16-C/EBPb can increase S100b transcription, with YAP/TAZ competing with C/EBPb for binding to PRDM16. This regulation by YAP/TAZ occurs by nuclear exclusion upon stimulation of cells by norepinephrine, thereby removing the competition with C/EBPb for binding PRDM16. The authors then did additional in vivo experiments showing that YAP/TAZ knockout mice are protected from weight gain and metabolic dysfunction with aging and high fat diet. They then show data suggesting knockout mice also have increased innervation and browning of visceral fat, which they propose is via S100b. Finally, they show data that S100b is decreased and TAZ levels are increased in humans with obesity.

Overall, this is an interesting manuscript and the authors are to be commended for the number of experiments they have done and for illustrating many of the key findings using multiple approaches. While the findings here will be of broad interest to researchers in this

field, a number of important issues need to be addressed to provide further confidence that the data support the conclusions made.

Major Points:

1) The whole mount imaging throughout the paper clearly shows increased sympathetic innervation, but at the magnification shown these are all large fibers that generally travel with blood vessels. In terms of effects on adipocyte thermogenesis, the key is to analyze the parenchymal filamentous arborizations which release catecholamines in the immediate vicinity of adipocytes. The images here lack the magnification or resolution to make out these nerve fibers. Addressing this point is central to providing a convincing argument that the phenotype in YAP/TAZ knockout mice is in fact due to increased innervation. In addition to higher power images, the authors could highlight the parenchymal innervation by showing z-stacks and videos of 3D reconstructions.

2) To follow on point 1, the data arguing for an effect of S100b on SNS innervation also needs to show clear views of the parenchymal fibers in the tissue. This is particularly important because while one paper has argued that a calsyntenin3b-S100b axis promotes sympathetic innervation (Xeng et al., Nature, 2019), a more recent paper (Qian et al, Nature, 2023) did not find this to be the case.

3) Since adipose tissue demonstrates regional architecture, for both the iWAT and eWAT studies, the authors need to clearly illustrate what portion of the tissue they are looking at and confirm that they always look at the same portion in each experiment.

4) Because the induction of UCP1 protein in iWAT seems out of proportion to the increase in transcripts and because no one has previously shown clear evidence of thermogenic adipocytes in eWAT, the authors should also perform at least some whole mount imaging with UCP1 antibodies in iWAT and eWAT. Moreover, their eWAT UCP1 Westerns should include a positive control, such as brown fat lysate.

Minor Points:

- 1) In Figure 1D, it isn't even clear that PRDM16 staining is nuclear. The authors need to include controls for non-specific signal.
- 2) For the Western blots, molecular weight markers should be shown next to the gels.
- 3) Why do the authors think that deletion of YAP/TAZ with UCP1 Cre results in this circling phenotype?
- 4) For the nuclear exclusion experiments in Figure 5, can the authors also show this by immunofluorescence imaging?

Reviewer #2 (Remarks to the Author):

Huang et al. explored the physiological role of YAP/TAZ in WAT adipocyte browning and sympathetic innervation using various cell based-, and mouse model-based experiments. The authors extensively discussed YAP/TAZ-PRDM16-C/EBP β -S100b axis in the context of adipocyte browning. They also highlighted the significance of YAP/TAZ activity in mature adipocytes for age-associated and diet-induced obesity. Notably, the deletion of YAP/TAZ showed promising effects on metabolic homeostasis through adipocyte browning. However, there are some critical concerns that should be addressed to improve the quality of the manuscript before publication.

Major:

1. In Fig.1, the authors use differentiated beige adipocytes to show PRDM16-YAP/TAZ complex. This may be interpreted as either the result or the cause of differentiation. Show the expression pattern and interaction pattern of PRDM16, CEBP, YAP, TAZ in each stage of adipogenesis and differentiation? Is the complex specifically formed before or after beige adipocyte differentiation?
2. TEAD is the major interacting partner of YAP/TAZ in various context including adipogenesis. The authors used TEAD binding-deficient mutants of YAP/TAZ in Fig. S4. Show whether these mutants, as well as YAP-5SA and TAZ-4SA can form complex with PRDM16

and also disrupt the PRDM16-CEBP complex.

3. The role of YAP/TAZ in the cell lineage and differentiation is known to be very critical. Though Fig. S2i and j present that adipocyte differentiation is not altered by YAP/TAZ deletion, it remains questionable what happens to the differentiation potential. Additionally, adipocyte browning is often accompanied by the downregulation of mature adipocyte markers. Authors need to clarify the cellular fate and differentiation potential upon YAP/TAZ KO. Result of PPAR γ , adiponectin, AP2 mRNA expression is insufficient to insist there is no impact on adipocyte differentiation.

Here are some references that present the role of Hippo pathway in adipogenesis: “Protein kinase A activates the Hippo pathway to modulate cell proliferation and differentiation, *Genes & Dev*, 2013”, TAZ is a negative regulator of PPAR γ activity in adipocytes and TAZ deletion improves insulin sensitivity and glucose tolerance, *Cell Metab*, 2020”, and “YAP as a key regulator of adipo-osteogenic differentiation in human MSCs, *Stem Cell Research & Therapy*, 2019”.

4. PRDM16, YAP/TAZ are also expressed and play critical roles in BATs. Although the authors mention the discrepancy a bit, there is a huge conflict in the results that cannot justify each other’s results regarding the effect of YAP/TAZ deletion in BAT using the same YT-UKO mice (*Cell Metab*. 2018 Mar 6;27(3):602-615.e4).

In line 402, authors propose a mechanism with no evidence. To provide in depth mechanism to support their data, authors should show that their mechanism does not take place in BAT. Validate the status of YAP/TAZ phosphorylation in WAT, beige WAT, and BAT, the expression and interaction of YAP/TAZ-PRDM16 in BAT, and whether YAP/TAZ can disrupt PRDM16-CEBP interaction in BAT.

5. Previous paper (*Nat Commun*. 2020 Oct 28;11(1):5455. doi: 10.1038/s41467-020-19229-3.) reports that loss of YAP/TAZ in adipocyte induces cell death and protects from obesity.

The authors use the same mouse model, however, does not describe cell death phenotype in their context. Instead, current data indicate that loss of YAP/TAZ promotes beige fat biogenesis in scWAT in an adipogenesis-independent manner. What happens to lipolytic activity? Could this contribute to the loss of lipid contents in YT-AKO? These discrepancies should be addressed by both additional experiments as well as in discussions.

6. Authors claim that NE-cAMP-PKA signaling induces YAP/TAZ phosphorylation. Further show whether PKA agonists can promote PRDM16-CEBP interaction.

7. Fig. 5c and 5d is hard to interpret. In 5c, NE and FSK increase YAP phosphorylation without affecting total YAP level. However, in 5d, cytoplasmic YAP shows no increase upon NE and FSK stimulation. Optimization of these experiments as well as blotting of p-YAP should be added in 5d. Also, perform IF to show the cytoplasmic translocation of YAP upon PKA activation.

8. Fig. 6c-e show severe loss of adipose tissue mass, and Fig. S6a-c show resistance to HFD-induced obesity from YT-AKO mice. These data present almost 50% reduction in lipid content in fat mass, however, only 13.7% increment in energy expenditure (VO_2) seems to be insufficient to generate such phenotype. Moreover, considering that no liver steatosis was observed in these mice, are the lipid metabolites accumulated in the blood samples?

9. Inflammatory response is considered one of the primary indicators for adipose tissue homeostasis and metabolic health condition. Suppression of inflammatory response was presented with RNA expression of TNF α , IL1b, and NOS2 in Fig. S5, 6, and 7, however, more detailed analysis should be required. Provide GSEA via RNA-seq of inflammatory response in YT-AKO WAT. Plus, it is widely accepted that macrophages contribute to adipocyte browning. Since Fig.6f indicates the presence of crown-like structures within the adipose tissue region, phenotyping result of macrophage populations would be very helpful.

10. Fig. 4c indicates a remarkable downregulation of CEBP α . Considering CEBP α is a crucial determinant for adipocyte property, provide explanation for S100b regulation via CEBP α .

11. Adipocyte-specific deletion of YAP/TAZ is a very artificial condition, and it requires further explanation about its relevance to human condition. Since this could be a potential therapeutic approach to improve metabolic abnormalities, it would be nice to suggest any possible therapeutic application to human health.

12. YAP/TAZ-PRDM16-C/EBP β -S100b axis has not been shown in age-associated and HFD-induced obese mice. Confirm this axis in your mouse models.

Minor:

1. Abbreviation for CCAAT/enhancer binding protein β should be 'C/EBP β '. Change every CEBP/ β to C/EBP β .

We thank the reviewers for their careful reading of our manuscript, and for their insightful and constructive comments for its improvement. Below we provide point-by-point detailed responses to the reviewers' inquiries. Our responses are presented in italic font.

We have highlighted all changes in blue in the manuscript.

REVIEWER COMMENTS

Reviewer #1 (Remarks to the Author):

In this manuscript, the authors investigated the role of YAP/TAZ in regulating the sympathetic innervation of adipose tissue via effects on S100b. They began by identifying YAP/TAZ as interacting with PRDM16 in adipocytes. Next, they made mice with adipocyte-specific deletion of YAP/TAZ and saw an increase in thermogenic beige fat, with associated metabolic benefits. Similar findings were reported with temporal deletion of YAP/TAZ in adipocytes using an AAV or a tamoxifen inducible system. Because in vitro differentiated adipocytes from YAP/TAZ knockout animals had no increase in thermogenic gene expression, the authors postulated that the in vivo phenotype was the result of a non-cell autonomous effect. So, they went on to investigate sympathetic innervation of adipose tissue. Their data suggests that YAP/TAZ deletion in adipocytes does result in increased SNS innervation, and profiling identified S100b as a possible mediator. Functional experiments were done to show that S100b can in fact promote sympathetic innervation. The authors then present a model, whereby PRDM16-C/EBP β can increase S100b transcription, with YAP/TAZ competing with C/EBP β for binding to PRDM16. This regulation by YAP/TAZ occurs by nuclear exclusion upon stimulation of cells by norepinephrine, thereby removing the competition with C/EBP β for binding PRDM16. The authors then did additional in vivo experiments showing that YAP/TAZ knockout mice are protected from weight gain and metabolic dysfunction with aging and high fat diet. They then show data suggesting knockout mice also have increased innervation and browning of visceral fat, which they propose is via S100b. Finally, they show data that S100b is decreased and TAZ levels are increased in humans with obesity.

Overall, this is an interesting manuscript and the authors are to be commended for the number of experiments they have done and for illustrating many of the key findings using multiple approaches. While the findings here will be of broad interest to researchers in this field, a number of important issues need to be addressed to provide further confidence that the data support the conclusions made.

We appreciate this reviewer for the positive comments.

Major Points:

1) The whole mount imaging throughout the paper clearly shows increased sympathetic innervation, but at the magnification shown these are all large fibers that generally travel with blood vessels. In terms of effects on adipocyte thermogenesis, the key is to analyze the parenchymal filamentous arborizations which release catecholamines in the immediate vicinity of adipocytes. The images here lack the magnification or resolution to make out these nerve fibers. Addressing this point is central to providing a convincing argument that the phenotype in YAP/TAZ knockout mice is in fact due to increased innervation. In addition to higher power images, the authors could highlight the parenchymal innervation by showing z-stacks and videos of 3D reconstructions.

Thank you for pointing out this issue. We have now shown new videos with high magnification and 3D reconstructions of parenchymal innervation of scWAT from YT-AKO and YT-FF mice (supplementary Movie 2). We calculated SNS density via 3D reconstruction of parenchymal innervation using Imaris software and performed statistical analyses. Consistently, Z-stack images of scWAT from YT-AKO mice showed much more parenchymal innervation compared to those of controls (Fig. R1a, or in Extended Data Fig. 2g and Fig. 3b). Besides, there was also more parenchymal innervation in eWAT of YT-AKO mice compared to controls (Fig. R1b, or in Extended Data Fig. 11b and Fig. 7h).

a

b

Fig. R1. YAP/TAZ loss promotes parenchymal innervation in adipose tissues. a, Z-stack images (200 μm) of Th staining of scWAT from adult YT-FF and YT-AKO mice. b, Z-stack images (400 μm) of Th staining of eWAT from adult YT-FF and YT-AKO mice. Scale bar 500 μm .

2) To follow on point 1, the data arguing for an effect of S100b on SNS innervation also needs to show clear views of the parenchymal fibers in the tissue. This is particularly important because while one paper has argued that a calsyntenin3b-S100b axis promotes sympathetic innervation (Xeng et al., Nature, 2019), a more recent paper (Qian et al, Nature, 2023) did not find this to be the case.

Indeed, Z-stack images of scWAT showed much denser parenchymal fibers when S100b was overexpressed in the tissue (Fig. R2, or in Extended Data Fig. 3h and Fig. 3j), which confirms the pro-innervation effect of S100b in adipose tissues. The inconsistency between the two Nature papers mainly lies in whether Calsyntenin 3b promotes sympathetic innervation in adipose tissues. The more recent paper (Nature 2023 Jan;613(7942):160-168) instead proposed that Calsyntenin 3b enforced adipocyte multilocularity, but the authors did not report whether S100b could regulate sympathetic innervation in adipose tissues.

Fig. R2. S100B overexpression promotes parenchymal innervation in scWAT. Z-stack images (200 μ m) of Th staining of scWAT from adult mice overexpressed with GFP or S100b. Scale bar 500 μ m.

3) Since adipose tissue demonstrates regional architecture, for both the iWAT and eWAT studies, the authors need to clearly illustrate what portion of the tissue they are looking at and confirm that they always look at the same portion in each experiment.

We do agree there are regional differences in adipose tissues. Thus, in order to avoid bias, we used whole adipose tissue but not certain portion of the tissue in the experiments like Western Blot, RT-qPCR, and whole mount imaging. We have now shown whole pictures for all the whole mount imaging.

4) Because the induction of UCP1 protein in iWAT seems out of proportion to the increase in transcripts and because no one has previously shown clear evidence of thermogenic adipocytes in eWAT, the authors should also perform at least some whole mount imaging with UCP1 antibodies in iWAT and eWAT. Moreover, their eWAT UCP1 Westerns should include a positive control, such as brown fat lysate.

As per the suggestion, we performed whole mount staining of UCP1 in both iWAT and eWAT. Compared to YT-FF controls, iWAT from adult YT-AKO mice under room temperature had more UCP1 signal (Fig. R3a, 3a1, 3b and 3b1, or in Extended Data Fig. 2h). As eWAT expresses extremely low UCP1 under room temperature, we performed UCP1 whole mount staining of eWAT after cold exposure for 7 days. The results revealed more browning area in eWAT of YT-AKO mice compared to YT-FF controls (Fig. R3c, 3c1, 3d and 3d1, or in Fig. 7i and supplementary Movie 3). In addition, we performed UCP1 Western blot of eWAT with brown fat and scWAT as positive controls (Fig. R3e, or in Extended Data Fig. 11a).

Fig. R3. YAP/TAZ deficiency promotes UCP1 expression in white adipose tissues. a, a1, b, b1, Whole mount imaging of UCP1 (a, b) and optical sections (a1, b1) in scWAT from adult YT-FF and YT-AKO mice under 22°C. c, c1, d, d1, Whole mount imaging of UCP1 (c, d) and optical sections (c1, d1) in eWAT from adult YT-FF and YT-AKO mice under 4°C for 7 days. e, Immunoblot of UCP1 of eWAT from mice under 4°C for 7 days, using scWAT and BAT lysate from mice under 22°C as positive controls.

Minor Points:

1) In Figure 1D, it isn't even clear that PRDM16 staining is nuclear. The authors need to include controls for non-specific signal.

To confirm the signal specificity of PRDM16 staining, we knocked out *Prdm16* in differentiated beige adipocytes derived from scWAT SVF and could not detect PRDM16

signal any more (Fig. R4, or in Extended Data Fig. 5f).

Fig. R4. Verification of PRDM16 staining specificity in differentiated beige adipocytes. Immunofluorescence (IF) of PRDM16 in differentiated beige adipocytes derived from scWAT SVF infected sequentially with Cas9 lentivirus and vec or Prdm16-gRNA AAV.

2) For the Western blots, molecular weight markers should be shown next to the gels.

We have now added molecular weight markers next to the Western blot gels.

3) Why do the authors think that deletion of YAP/TAZ with UCP1 Cre results in this circling phenotype?

We think it is probably due to a leakage of $UCP1^{Cre}$ expression in the brain. Using $UCP1^{Cre};tdTomato$ reporter mice, Kristin et al. recently revealed that $UCP1^{Cre}$ is not restricted to the thermogenic adipocytes as assumed but can express throughout the brain (Mol Metab. 2022 Jan;55:101405). Besides, Willem et al. also found central nervous system expresses UCP1 in squirrels (Proc Natl Acad Sci USA. 2015 Feb 3;112(5):1607-12.). Thus, Yap/Taz might be deleted in the central nervous system of YF-UKO mice. Since YAP/TAZ are important for the development of brain like cerebellum, which is critical for locomotor coordination (Dev Dyn. 2022 Oct;251(10):1644-1665), loss of YAP/TAZ may negatively affect brain development and cause abnormal behaviors such as circling phenotype in this study.

4) For the nuclear exclusion experiments in Figure 5, can the authors also show this by immunofluorescence imaging?

We performed immunofluorescence (IF) imaging in differentiated beige adipocytes derived from SVF of scWAT. Consistent with Figure 5, norepinephrine (NE) and forskolin (FSK) treatment significantly decreased YAP/TAZ level in the nucleus, and the decrease by NE was rescued by PKA inhibitor H89 (Fig. R5a-d, or in Fig. 5d-g).

a

b

Fig. R5. NE- β -AR-PKA signaling promotes YAP/TAZ extrusion from the nucleus. a, b, YAP, PRDM16 IF in differentiated beige adipocytes derived from scWAT SVF and treated with indicated reagents for 30 minutes (a) and quantification of IF intensity of YAP and PRDM16 in the nucleus (b). c, d, TAZ, PRDM16 IF in differentiated beige adipocytes derived from scWAT SVF and treated with indicated reagents for 30 minutes (c) and quantification of IF intensity of TAZ in the nucleus (d).

Reviewer #2 (Remarks to the Author):

Huang et al. explored the physiological role of YAP/TAZ in WAT adipocyte browning and sympathetic innervation using various cell based-, and mouse model-based experiments. The authors extensively discussed YAP/TAZ-PRDM16-C/EBP β -S100b axis in the context of adipocyte browning. They also highlighted the significance of YAP/TAZ activity in mature adipocytes for age-associated and diet-induced obesity. Notably, the deletion of YAP/TAZ showed promising effects on metabolic homeostasis through adipocyte browning. However, there are some critical concerns that should be addressed to improve the quality of the manuscript before publication.

Major:

1. In Fig.1, the authors use differentiated beige adipocytes to show PRDM16-YAP/TAZ complex. This may be interpreted as either the result or the cause of differentiation. Show the expression pattern and interaction pattern of PRDM16, CEBP, YAP, TAZ in each stage of adipogenesis and differentiation? Is the complex specifically formed before or after beige adipocyte differentiation?

As the focus of this study is the role of YAP/TAZ in adipocytes and we deleted them by Adiponectin^{Cre} whose expression is restricted to mature adipocytes but not progenitors or preadipocytes, we think there is no direct relevance with adipocyte differentiation. As per the reviewer's suggestion, we examined the expression of PRDM16, C/EBP β , YAP and TAZ in preadipocytes and differentiated beige adipocytes and found that PRDM16 was almost only expressed in differentiated adipocytes. Also, YAP and TAZ, especially YAP, were sharply upregulated in mature beige adipocytes (Fig. R6, or in Extended Data Fig. 4p). These results indicate that PRDM16-YAP/TAZ interaction mainly happens in differentiated beige adipocytes.

Fig. R6. The expression pattern of indicated proteins in preadipocytes and beige adipocytes. Immunoblot and quantification of indicated proteins in scWAT-derived SVF before and after differentiation (n=3).

2. TEAD is the major interacting partner of YAP/TAZ in various context including adipogenesis. The authors used TEAD binding-deficient mutants of YAP/TAZ in Fig. S4. Show whether these mutants, as well as YAP-5SA and TAZ-4SA can form complex with PRDM16 and also disrupt the PRDM16-CEBP complex.

As per the reviewer's request, we performed additional co-IP assays and found that the TEAD binding-deficient mutants of YAP/TAZ (YAP-S79A, TAZ-S51A) could also disrupt PRDM6-C/EBP β interaction to similar extent as wild-type YAP and TAZ (Fig. R7a and 7b, or in Extended Data Fig. 5b and 5c). Moreover, the active forms of YAP/TAZ (YAP-5SA, TAZ-4SA), which mainly located in the nucleus, could block PRDM16-C/EBP β complex more robustly than wild-type YAP and TAZ (Fig. R7c and 7d, or in Extended Data Fig. 6a and 6b).

Fig. R7. Both active and TEAD-binding mutated YAP/TAZ disrupt the PRDM16-C/EBP β interaction. Co-IP of C/EBP β and PRDM16 in HEK-293T cells expressed with indicated proteins (a, c) and quantification of immunoprecipitated C/EBP β (b, d).

3. The role of YAP/TAZ in the cell lineage and differentiation is known to be very critical. Though Fig. S2i and j present that adipocyte differentiation is not altered by YAP/TAZ deletion, it remains questionable what happens to the differentiation potential. Additionally, adipocyte browning is often accompanied by the downregulation of mature adipocyte markers. Authors need to clarify the cellular fate and differentiation potential upon YAP/TAZ KO. Result of PPAR γ , adiponectin, AP2 mRNA expression is insufficient to insist there is no impact on adipocyte differentiation.

Here are some references that present the role of Hippo pathway in adipogenesis: “Protein kinase A activates the Hippo pathway to modulate cell proliferation and differentiation, *Genes & Dev*, 2013”, TAZ is a negative regulator of PPAR γ activity in adipocytes and TAZ deletion improves insulin sensitivity and glucose tolerance, *Cell Metab*, 2020”, and “YAP as a key regulator of adipo-osteogenic differentiation in human MSCs, *Stem Cell Research & Therapy*, 2019”.

We would like to clarify that we used Adiponectin-promoter-driven AAV to specifically overexpress S100b in mature adipocytes and showed that S100b did not alter adipocyte identity, having no direct relevance with differentiation (now Extended Data Fig. 3e, 3f, or Fig S2i, 2j as the reviewer cited).

*As the reviewer pointed out, YAP/TAZ are very important regulators for preadipocyte proliferation and differentiation. However, most of these studies are performed in preadipocytes (*Genes Dev*. 2013 Jun 1;27(11):1223-32; *Stem Cell Res Ther*. 2019 Dec 18;10(1):402; *Science*. 2005 Aug 12;309(5737):1074-8). In this study, we focus on the physiological roles of YAP/TAZ in mature adipocytes and we used Adiponectin^{Cre} to manipulate mature adipocytes. We found that Yap/Taz deletion in adipocytes did not alter their identity as revealed by adipocyte markers like Adipoq, Ap2 and PPAR γ . To further support this point, we deleted Yap/Taz in differentiated beige adipocytes derived from SVF of scWAT from Cas9^{tg} mice. The Oil-Red O staining results showed that YAP/TAZ deficiency in differentiated beige adipocytes did not alter adipocyte identity (Fig. R8a, 8b, or in Extended Data Fig. 2e and 2f).*

Fig. R8. YAP/TAZ deletion in differentiated beige adipocytes does not alter adipocyte identity. a, b, Oil Red O staining (a) and quantification (b) of beige adipocytes derived from Cas9^{tg/tg} scWAT SVF and infected with vec or Yap/Taz-gRNA (YT-gRNA) AAV as used in Extended Data Fig. 2c (n=3). Scale bar, 100 μ m.

4. PRDM16, YAP/TAZ are also expressed and play critical roles in BATs. Although the authors mention the discrepancy a bit, there is a huge conflict in the results that cannot justify each other's results regarding the effect of YAP/TAZ deletion in BAT using the same YT-UKO mice (Cell Metab. 2018 Mar 6;27(3):602-615.e4).

In line 402, authors propose a mechanism with no evidence. To provide in depth mechanism to support their data, authors should show that their mechanism does not take place in BAT. Validate the status of YAP/TAZ phosphorylation in WAT, beige WAT, and BAT, the expression and interaction of YAP/TAZ-PRDM16 in BAT, and whether YAP/TAZ can disrupt PRDM16-CEBP interaction in BAT.

In figure 7a-c, we showed that there was less total but more phosphorylated YAP/TAZ in BAT compared to scWAT and eWAT. Following the reviewer's suggestion, we have now performed additional endogenous co-IP assays both in BAT and scWAT from YT-AKO and YT-FF mice. Consistent with our hypothesis, YAP/TAZ loss in scWAT markedly promoted PRDM16-C/EBP β interaction while there was no effect in BAT (Fig. R9a and R9b, or in Fig. 4j and Extended Data Fig. 4m).

Fig. R9. YAP/TAZ loss promotes PRDM16-C/EBPβ interaction in scWAT but not BAT. a, b, endogenous co-IP of PRDM16 and C/EBPβ and quantification of immunoprecipitated PRDM16 in scWAT (a) and BAT (b) from YT-FF and YT-AKO mice (n=3).

5. Previous paper (Nat Commun. 2020 Oct 28;11(1):5455. doi: 10.1038/s41467-020-19229-3.) reports that loss of YAP/TAZ in adipocyte induces cell death and protects from obesity. The authors use the same mouse model, however, does not describe cell death phenotype in their context. Instead, current data indicate that loss of YAP/TAZ promotes beige fat biogenesis in scWAT in an adipogenesis-independent manner. What happens to lipolytic activity? Could this contribute to the loss of lipid contents in YT-AKO? These discrepancies should be addressed by both additional experiments as well as in discussions.

In this study, we found that YAP/TAZ loss in white adipose tissues can promote browning,

which increases the burning of lipid contents and transforms the stored energy into heat, leading to less lipid contents in YT-AKO mice. To evaluate lipolytic activity, we examined the expression of hormone-sensitive lipase (HSL) and Perilipin1 and their phosphorylation status in scWAT and eWAT of HFD-induced obese mice and aging mice respectively. The Western blot results showed that induced deletion of Yap/Taz in adipose tissues did not affect the expression of HSL and Perilipin1 but slightly increased pHSL in scWAT and pPerilipin1 in eWAT from YT-iAKO mice under HFD condition (Fig. R10a and 10b, or in Extended Data Fig. 9d-g). Under aging condition, there was no effect on the expression and phosphorylation level of HSL and Perilipin1 in scWAT and eWAT from YT-AKO mice (Fig. R10c and 10d, or in Extended Data Fig. 7c-f). These data indicate that YAP/TAZ loss promotes lipid loss mainly by enhancing browning but not lipolysis in adipose tissues.

We have added this point in Discussion as “Wang et al. reported that YAP/TAZ can protect against white adipocyte death and YAP/TAZ loss in adipocytes induces cell apoptosis and impedes fat mass accumulation under HFD feeding condition. However, they did not explain where to go and how to be utilized of released lipids from dead cells. Since lipolytic ability of WAT from YT-AKO did not enhance markedly, our data suggest that the released lipids from apoptotic adipocytes are utilized as fuels for increased beige fat.”.

a

b

Fig. R10. YAP/TAZ loss does not significantly affect lipolytic ability of adipose tissues under HFD and aging condition. a, b, Immunoblot and quantification of indicated proteins in scWAT (a) and eWAT (b) from YT-iAKO or control mice fed HFD. c, d, Immunoblot and quantification of indicated proteins in scWAT (c) and eWAT (d) from YT-AKO or control aging mice. N=3.

6. Authors claim that NE-cAMP-PKA signaling induces YAP/TAZ phosphorylation. Further show whether PKA agonists can promote PRDM16-CEBP interaction.

We have now shown that NE and PKA agonist (FSK) could significantly promote PRDM16-C/EBP β interaction (Fig. R11a, or in Fig. 5h).

Fig. R11. NE-cAMP-PKA signaling promotes PRDM16-C/EBP β interaction. Endogenous co-IP of PRDM16 and C/EBP β and quantification of immunoprecipitated PRDM16 in SVF-derived beige adipocytes treated with indicated reagents for 30 minutes (n=3).

7. Fig. 5c and 5d is hard to interpret. In 5c, NE and FSK increase YAP phosphorylation without affecting total YAP level. However, in 5d, cytoplasmic YAP shows no increase upon NE and FSK stimulation. Optimization of these experiments as well as blotting of p-YAP should be added in 5d. Also, perform IF to show the cytoplasmic translocation of YAP upon PKA activation.

When YAP/TAZ are phosphorylated, they will be quickly translocated to the cytoplasm and followed by degradation. In previous Fig 5d, the nucleus-cytoplasm separation procedure took about 2 hours, during which the phosphorylated YAP/TAZ was degraded and thus we could not see an increase of cytoplasmic YAP/TAZ. In contrast, the cell samples in Fig 5c took about 15 minutes to harvest and lyse, which avoided massive YAP/TAZ degradation and thus did not significantly affect total YAP/TAZ level. As per the reviewer's suggestion, we have now added p-YAP/TAZ in previous Fig. 5d, showing that NE or FSK treatment significantly increased p-YAP/TAZ in the cytoplasm (Fig. R12a, or in Extended Data Fig. 5g and i).

Following the reviewer #1's and #2's suggestion, we performed immunofluorescence (IF) imaging in differentiated beige adipocytes derived from SVF of scWAT. Consistent with Figure 5, norepinephrine (NE) and forskolin (FSK) treatment significantly decreased YAP/TAZ level in the nucleus, and the decrease by NE was rescued by PKA inhibitor H89 (Fig. R12b-e, or in Fig. 5d-g).

a

b

c

Fig. R12. NE- β AR-PKA signaling promotes YAP/TAZ extrusion from the nucleus. *a*, Immunoblot of indicated proteins in cytoplasm and nucleus fractions after treated with indicated reagents for 30 minutes. *b-e*, YAP, TAZ and PRDM16 IF in differentiated beige adipocytes derived from scWAT SVF and treated with indicated reagents for 30 minutes (*b*, *d*) and quantification of IF intensity of indicated proteins in the nucleus (*c*, *e*).

8. Fig. 6c-e show severe loss of adipose tissue mass, and Fig. S6a-c show resistance to HFD-induced obesity from YT-AKO mice. These data present almost 50% reduction in lipid content in fat mass, however, only 13.7% increment in energy expenditure (VO₂) seems to be insufficient to generate such phenotype. Moreover, considering that no liver steatosis was observed in these mice, are the lipid metabolites accumulated in the blood samples?

As mentioned in the response to point 5, YAP/TAZ loss in white adipose tissues can promote their browning, which increases the burning of lipid contents and transforms the stored energy into heat. Noteworthy, it is a continual process. Thus, daily 13.7% increment in energy expenditure is sufficient to reduce 50% of lipid content in fat mass of aging mice as time goes on. As per the reviewer's suggestion, we examined triglyceride (TG) and non-esterified fatty acid (NEFA) in the sera of YT-AKO, YT-iAKO and respective controls fed HFD, and of aging YT-AKO and YT-FF mice. There was no lipid metabolites accumulation in the sera of YT-AKO and YT-iAKO mice under HFD or aging condition, suggesting that the lost fat was burned by beige fat (Fig. R13a-c, or in Extended Data Fig. 7i, 7j, 8l, 8m, 9n and 9o).

Fig. 13. Lipid metabolites do not accumulate in the sera of YT-AKO and YT-iAKO mice

under HFD feeding or aging condition. a, NEFA and TG concentration in the sera of YT-FF and YT-AKO aging mice (n=5). b, NEFA and TG concentration in the sera of YT-FF and YT-AKO mice fed HFD for 15 weeks (n=6). c, NEFA and TG concentration in the sera of YT-FF and YT-iAKO mice fed HFD for 20 weeks (n=6).

9. Inflammatory response is considered one of the primary indicators for adipose tissue homeostasis and metabolic health condition. Suppression of inflammatory response was presented with RNA expression of TNF α , IL1 β , and NOS2 in Fig. S5, 6, and 7, however, more detailed analysis should be required. Provide GSEA via RNA-seq of inflammatory response in YT-AKO WAT. Plus, it is widely accepted that macrophages contribute to adipocyte browning. Since Fig.6f indicates the presence of crown-like structures within the adipose tissue region, phenotyping result of macrophage populations would be very helpful.

To dissect the inflammatory response of scWAT from YT-AKO and control mice, we analyzed the RNA-seq data of scWAT from 8-week-old mice fed normal diet. GSEA of acute and chronic inflammatory responses did not show significant difference between YT-AKO and YT-FF mice, indicating that YAP/TAZ deletion does not influence inflammatory response but promote browning of scWAT under normal diet feeding condition (Fig. R14a, or in Extended Data Fig. 10a and 10b). On the other hand, WAT is infiltrated with inflammatory macrophages and consequently associated with systemic metabolic abnormalities under HFD-induced obesity condition (*Am J Physiol Cell Physiol.* 2021 Mar 1;320(3):C375-C391). To determine whether YAP/TAZ deletion-induced energy expenditure increase and fat loss affected inflammatory response, we analyzed macrophage populations in WAT from YT-iAKO mice fed HFD by flow cytometry. Consistent with the qPCR results, macrophages particularly pro-inflammatory M1 macrophages were significantly less infiltrated into scWAT and eWAT (Fig. R14b-d, or in Extended Data Fig. 10c and 10h-k). A previous study reported that percentage of M2 adipose tissue macrophages was decreased in old mice and the old fat shifted to a proinflammatory environment (*J Immunol.* 2011 Dec 15;187(12):6208-16). Consistently, we have now shown that percentages of M2 macrophages were significantly increased in scWAT and eWAT from YT-AKO aging mice, indicating an anti-inflammatory microenvironment in YT-AKO adipose tissues (Fig. R14e and 14f, or in Extended Data Fig. 10d-g).

a

b

c

d

e

f

Fig. R14. YAP/TAZ loss reduces inflammatory response in scWAT and eWAT in HFD or aging condition. a, GSEA of RNA-seq data of scWAT from adult YT-FF and YT-AKO mice ($n=3$). b, Gating strategy of flow cytometry for macrophage phenotyping. c, d, Cell numbers and percentages of indicated cell types in scWAT (c) and eWAT (d) of YT-FF and YT-iAKO mice fed HFD for 20 weeks ($n=6$). e, f, Cell numbers and percentages of indicated cell types in scWAT (e) and eWAT (f) of aging mice ($n=6$).

10. Fig. 4c indicates a remarkable downregulation of CEBP α . Considering CEBP α is a crucial determinant for adipocyte property, provide explanation for S100b regulation via CEBP α .

Although C/EBP α was predicted to bind to the S100b promoter (SP), analyses of the C/EBP α ChIP-seq data (GSM1218859) in brown adipocytes revealed barely detectable binding signal on the SP compared to C/EBP β (Fig. R15a, or in Fig. 4d and Extended Data Fig. 4h). On the other hand, Yutaka et al. reported that PRDM16 can interact with C/EBP α according to the MS/MS results (Cell Metab. 2018 Jan 9;27(1):180-194.e6). Thus, overexpressed C/EBP α might compete with C/EBP β for binding to PRDM16, which blocked the formation of C/EBP β -PRDM16 complex and consequently inhibited S100b expression as shown in Fig. 4c.

Fig. R15. C/EBP α barely binds to the S100b promoter compared to C/EBP β . Alignment of C/EBP α and C/EBP β ChIP-seq peaks at the S100b gene locus.

11. Adipocyte-specific deletion of YAP/TAZ is a very artificial condition, and it requires further explanation about its relevance to human condition. Since this could be a potential therapeutic approach to improve metabolic abnormalities, it would be nice to suggest any possible therapeutic application to human health.

By adipocyte-specific deletion of YAP/TAZ, we prove in concept that inhibiting YAP/TAZ in adipose tissues particularly in visceral white adipose tissue could be an effective therapeutic approach to treat diet-induced or aging-related obesity and their associated metabolic disorders. Furthermore, Hippo signaling inactivation is associated with various cancers and high nucleus levels of YAP/TAZ expression correlate with poor prognosis and therapeutic resistance (Nat Rev Drug Discov. 2020 Jul;19(7):480-494). Thus, targeting the Hippo pathway particularly YAP/TAZ could not only improve metabolism abnormalities but also enhance therapeutic effects in some aggressive cancers. RNA interference (RNAi) has been demonstrated to be a powerful therapeutic strategy and the advancements in pharmaceutical chemistry bring RNAi-based drugs into clinical application (Nat Rev Genet. 2022 May;23(5):265-280). Besides, proteolysis targeting chimeras (PROTACs) technology is an emerging technology, which can effectively promote degradation of proteins that are difficult to target with small molecules (Nat Rev Drug Discov. 2022 Mar;21(3):181-200). All these technologies provide promising means to target YAP/TAZ for the prevention and

treatment of obesity and associated metabolic disorders and cancers. We have now added this point in the discussion section.

12. YAP/TAZ-PRDM16-C/EBP β -S100b axis has not been shown in age-associated and HFD-induced obese mice. Confirm this axis in your mouse models.

We demonstrated that YAP/TAZ could disrupt PRDM16-C/EBP β complex by competing with C/EBP β to bind PRDM16, which subsequently inhibited S100b transcription in 293T cell line, differentiated beige adipocytes and scWAT by co-IP, Cut&Tag and RT-qPCR assays (Fig. 4 and Fig R9). We also demonstrated that deleting YAP/TAZ specifically in adipocytes could promote browning of scWAT of adult mice in a S100b-dependent manner (Fig. 3). Consistent with these mechanistic discoveries, we revealed that YAP/TAZ loss in adipocytes greatly improved obesity-related metabolic dysfunction by upregulation of browning program both in aging-associated and HFD-induced obese mouse models (Fig 6 and Extended data Fig. 7 to 9). We further confirmed the YAP/TAZ-PRDM16-C/EBP β -S100b axis in these two models by showing that S100b expression was robustly increased upon adipocyte-specific deletion of YAP/TAZ. We have now shown that there was no alteration of PRDM16 and C/EBP β in these two models (Fig. R16a-d), consistent with the correlation data of obese and old individuals (Fig. 8a-e and Extended Data Fig. 12a-e). We have incorporated all these data into the new figures (Fig. 6j, Extended Data Fig. 8h, 11c and 11d).

Fig. R16. YAP/TAZ deletion promotes S100b expression in adipose tissues from aging

and HFD-induced obese mice. a, b, qPCR analyses of scWAT (a) and eWAT (b) from aging mice (n=5). b, qPCR analyses of scWAT (c) and eWAT (d) from HFD-induced obese mice (n=6).

Minor:

1. Abbreviation for CCAAT/enhancer binding protein β should be 'C/EBP β '.
Change every CEBP/ β to C/EBP β .

Thank you for pointing this out. We have now changed CEBP/ β to C/EBP β .

REVIEWERS' COMMENTS

Reviewer #1 (Remarks to the Author):

In this revised manuscript, the authors have included a number of new experiments that have strengthened the overall manuscript. The concerns I raised in my initial review have now been largely addressed. One point which would be nice to clarify is the UCP1 whole mount imaging in Figure R3 of the rebuttal and Extended Figure 2 of the manuscript. The UCP1 staining appears to show high background, especially with the strong staining in the lymph node. To resolve this possibility, the authors could do a control experiment where they stain with secondary antibody alone.

Reviewer #2 (Remarks to the Author):

The authors provided satisfied additional data to address the concerns raised.

Reviewer #1 (Remarks to the Author):

In this revised manuscript, the authors have included a number of new experiments that have strengthened the overall manuscript. The concerns I raised in my initial review have now been largely addressed. One point which would be nice to clarify is the UCP1 whole mount imaging in Figure R3 of the rebuttal and Extended Figure 2 of the manuscript. The UCP1 staining appears to show high background, especially with the strong staining in the lymph node. To resolve this possibility, the authors could do a control experiment where they stain with secondary antibody alone.

As the reviewer pointed out, the lymph nodes of scWAT in Extended Figure 2 showed high signal background in the lymph nodes, which was also reported by Jingyi Chi et al in their whole mount imaging of scWAT (Cell Metab. 2018 Jan 9;27(1):226-236.e3). They found that the lymph nodes of cleared scWAT via their method showed strong green autofluorescence (Fig. R1). Since we applied the same method, we think the proposed control experiment with secondary antibody alone is not suitable to resolve this autofluorescence issue. Actually, we observed significantly increased UCP1 signal in the dorsolumbar region of scWAT from YT-AKO mice, while it was barely detectable in that of YT-FF control mice. Combined with the UCP1 western blot and H&E results in Figure 2d and 2e, these data demonstrate that scWAT from YT-AKO mice have more beige cells and higher level of UCP1 especially in the dorsolumbar region. To let readers aware of this issue, we add a sentence in the method as "Notably, compared to stronger TH signal in scWAT, UCP1 signal is weaker and thus the background signal derived from autofluorescence especially in the lymph nodes of scWAT is obvious."

[FIGURE REDACTED]

Fig. R1. Tissue autofluorescence of whole mount imaging of scWAT (green channel).